# Targeting the Notch-regulated non-coding RNA *TUG1* for glioma treatment

Keisuke Katsushima[1], Atsushi Natsume[2], Fumiharu Ohka[1,2], Keiko Shinjo[1], Akira Hatanaka[1], Norihisa Ichimura[1], Shinya Sato[3], Satoru Takahashi[3], Hiroshi Kimura[4], Yasushi Totoki[5], Tatsuhiro Shibata[5,6], Mitsuru Naito[7], Hyun Jin Kim[7], Kanjiro Miyata[7,8], Kazunori Kataoka[7,8,9] & Yutaka Kondo[1,10]

Targeting self-renewal is an important goal in cancer therapy and recent studies have focused on Notch signalling in the maintenance of stemness of glioma stem cells (GSCs). Understanding cancer-specific Notch regulation would improve specificity of targeting this pathway. In this study, we find that Notch1 activation in GSCs specifically induces expression of the lncRNA, *TUG1*. *TUG1* coordinately promotes self-renewal by sponging miR-145 in the cytoplasm and recruiting polycomb to repress differentiation genes by locus-specific methylation of histone H3K27 via YY1-binding activity in the nucleus. Furthermore, intravenous treatment with antisense oligonucleotides targeting *TUG1* coupled with a drug delivery system induces GSC differentiation and efficiently represses GSC growth *in vivo*. Our results highlight the importance of the Notch-lncRNA axis in regulating self-renewal of glioma cells and provide a strong rationale for targeting *TUG1* as a specific and potent therapeutic approach to eliminate the GSC population.

[1] Department of Epigenomics, Nagoya City University Graduate School of Medical Sciences, Nagoya 467-8601, Japan. [2] Department of Neurosurgery, Nagoya University School of Medicine, Nagoya 466-8550, Japan. [3] Experimental Pathology and Tumor Biology, Nagoya City University Graduate School of Medical Sciences, Nagoya 467-8601, Japan. [4] Cell Biology Unit, Institute of Innovative Research, Tokyo Institute of Technology, Tokyo 226-8501, Japan. [5] Division of Cancer Genomics, National Cancer Center, Tokyo 104-0045, Japan. [6] Laboratory of Molecular Medicine, Human Genome Center, The Institute of Medical Science, The University of Tokyo, Tokyo 108-8639, Japan. [7] Center for Disease Biology and Integrative Medicine, Graduate School of Medicine, The University of Tokyo, Tokyo 113-0033, Japan. [8] Department of Materials Engineering, Graduate School of Engineering, The University of Tokyo, Tokyo 113-8656 Japan. [9] Innovation Center of Nanomedicine, Kawasaki Institute of Industry Promotion, Kawasaki 210-0821, Japan. [10] Precursory Research for Embryonic Science and Technology (PRESTO), Japan Science and Technology Agency, Tokyo 102-8666, Japan. Correspondence and requests for materials should be addressed to Y.K. (email: ykondo@med.nagoya-cu.ac.jp).

Glioblastomas (GBMs) show heterogeneous histological features. These distinct phenotypes are thought to be due to their derivation from glioma stem cells (GSCs), which are a highly tumorigenic and self-renewing sub-population of tumour cells that have different functional characteristics[1,2]. As the self-renewing capability and multipotency may be closely associated with tumour progression and invasion, understanding the mechanisms underlying the regulation of stemness properties of tumour cells has been actively investigated[3–5].

In the neural stem cell, the Notch signalling pathway has a dominant role in inhibiting differentiation through the activities of its downstream effectors, such as Hairy and enhancer of split 1/5 (Hes1/5), which repress the implementation of neurogenic programs[6]. In the context of oncogenesis, Notch signalling has been shown to promote GSC self-renewal and to suppress GSC differentiation[7]. The mechanisms by which Notch regulates brain tumour stem cells appear to be similar to those governing regulation of neural stem cells during neural development[8]. However, the mechanism by which Notch signalling and its downstream effectors maintains the stemness properties of GSCs through the function of a certain set of genes, such as SOX2, MYC and Nestin, remains unresolved.

It has been increasingly demonstrated that large parts of the human genome are transcribed into non-coding RNAs (ncRNAs). A recent study showed that Notch-triggered oncogenic activity can be due to not only its ability to regulate proteins, but also long ncRNAs (lncRNAs) in leukaemia[9]. To date, studies have shown that lncRNAs affect the chromatin structure, mRNA stability and miRNA-mediated gene regulation by acting as competing endogenous RNA or natural miRNA sponges[10–12].

Among these lncRNAs, Taurine Upregulated Gene 1 (TUG1) has been reported as cancer-related, and can bind to polycomb repressive complex 2 (PRC2) or PRC1 as well as repress gene expression[13,14]. TUG1 was originally identified as a transcript upregulated by taurine, whose function is associated with retinal development[15]. In the context of human malignancies, it is overexpressed in bladder cancer, gastric cancer and osteosarcoma[16–18], whereas it is downregulated in non-small cell lung cancer[19], suggesting context-dependent roles in different types of cancers. As the length of the TUG1 lncRNA is not short, ~7.1 kb, it is plausible that TUG1 has multiple functions, which remain unknown.

Here, we show that TUG1, the expression of which is regulated by the Notch signalling pathway, was highly expressed in GSCs and maintained the stemness features of glioma cells. Furthermore, we developed new antisense oligonucleotides (ASO) targeting TUG1 coupled with a potent drug delivery system (DDS), which can be used intravenously to provide efficient and selective delivery to glioma cells at sufficient concentrations to acquire antitumour effects[20]. Our observations indicate that Notch-directed TUG1 is an effective epigenetic modulator that regulates the cancer stem cell population.

## Results

**Identification of TUG1 as a Notch-regulated lncRNA.** The two GSC populations assessed (1228-GSC and 222-GSC) showed high levels of GSC markers, such as CD15 and SOX2, and GSC characteristics by functional analyses (in vivo limiting dilution assay) (Supplementary Fig. 1), which agree with the previously described model[1,21]. To identify lncRNAs that are regulated by Notch signalling, we first performed RNA-sequencing (RNA-seq) analysis to investigate the lncRNA expression profile in these two GSC populations[5,22]. In Notch signalling, the Notch intracellular domain (NICD) translocates into the nucleus and binds the transcription factor, RBPJκ[23]. Therefore, we selected lncRNAs that contain the binding motifs of RBPJκ (-TTCCCAG/C-) around their transcriptional start site (TSS, $-2$ kb to $+1$ kb, $-$ indicates upstream of TSS and $+$ indicates downstream of TSS) using the JASPAR CORE database (http://jaspar.binf.ku.dk/)[24]. Commonly highly expressed lncRNAs with associated RBPJκ motifs in both GSC populations were identified (Top 50, Fig. 1a).

To identify lncRNA potentially involved in Notch-induced GSC maintenance, we further performed lncRNA microarray analysis in GSCs with either treatment by small interfering RNA (siRNA) targeting Notch1 (si-Notch1), si-JAG1 (Notch ligand)[25], or γ-secretase inhibitors (N-S-phenyl-glycine-t-butyl ester (DAPT) and RO4929097, which is undergoing clinical trial in glioma) (Supplementary Data 1). Notably, the use of DAPT or RO4929097 may not be very specific to Notch signalling inhibition but rather inhibition of γ-secretase. Therefore, we also examined lncRNA expression in si-JAG1 and si-Notch1 treated GSCs by RNA-seq analysis in addition to lncRNA microarray analysis and found that results of both analyses were quite concordant ($R^2 = 0.9014$ and 0.9152 in 1228- and 222-GSCs, respectively) (Supplementary Fig. 2 and Supplementary Data 2). These analyses revealed that six lncRNAs were commonly downregulated in the two GSC populations following inhibition of Notch signalling (Fig. 1b, Supplementary Fig. 3a–d and Supplementary Data 3). Among these, TUG1 (NR_002323.1) was the only lncRNA, which was also highly expressed as determined by RNA-seq analysis and possessed multiple RBPJκ motifs around its promoter region (Fig. 1a,c). Indeed, Notch1 bound to RBPJκ motifs within the upstream region of TUG1 TSS (Fig. 1d). We confirmed that level of TUG1 expression was efficiently reduced in GSCs following treatment with either si-Notch1, si-JAG1 or γ-secretase inhibitors (DAPT and RO4929097) (Fig. 1e–g and Supplementary Fig. 3a–d). Compellingly, downregulation of TUG1 expression by Notch signalling inhibition was rescued by ectopic NICD expression (Supplementary Fig. 3e–g). These data indicate that Notch signalling predominantly regulates TUG1 expression in glioma cells.

**TUG1 maintains stemness features of GSCs.** Inhibition of TUG1 by siRNA in GSCs reduced cell proliferation via induction of apoptosis as detected by FACS analysis with annexin V and 7-aminoactinomycin D staining (Fig. 2a,b). Notch signalling induces the promoter activity of Nestin, a well-characterized marker of stemness features[26]. We analysed the effect of TUG1 depletion in GSC lines that stably expressed enhanced GFP (EGFP) under the regulation of the Nestin promoter (GSC-pE-Nes;[5,27]). In GSC-pE-Nes neurospheres, ~80% of cells were positive for Nestin-EGFP. The majority of Nestin-EGFP-expressing cells were positive for Notch1, whereas Nestin-EGFP-negative cells were positive for JAG1 and lacked Notch1 expression (Supplementary Fig. 4a). Although the viability of both EGFP-positive and EGFP-negative cells were comparable, Nestin-EGFP-positive cells were highly tumorigenic compared with Nestin-EGFP-negative cells (Supplementary Fig. 4b,c). TUG1 depletion in GSC lines reduced Nestin promoter activity together with impaired neurosphere formation in GSC-pE-Nes (Fig. 2c,d). These morphological and functional changes were also observed after Notch1 and JAG1 inhibition (Supplementary Fig. 4d,e). TUG1 depletion also effectively decreased the expression of stemness-associated genes (SOX2, MYC, Nestin and CD15) in two GSC lines (Fig. 2e,f). Taken together, it is evident that Notch-regulated TUG1 affects the stemness features of GSCs.

**TUG1 antagonizes miR-145 and regulates SOX2 and MYC.** Some lncRNAs are known to act as miRNA sponges in the

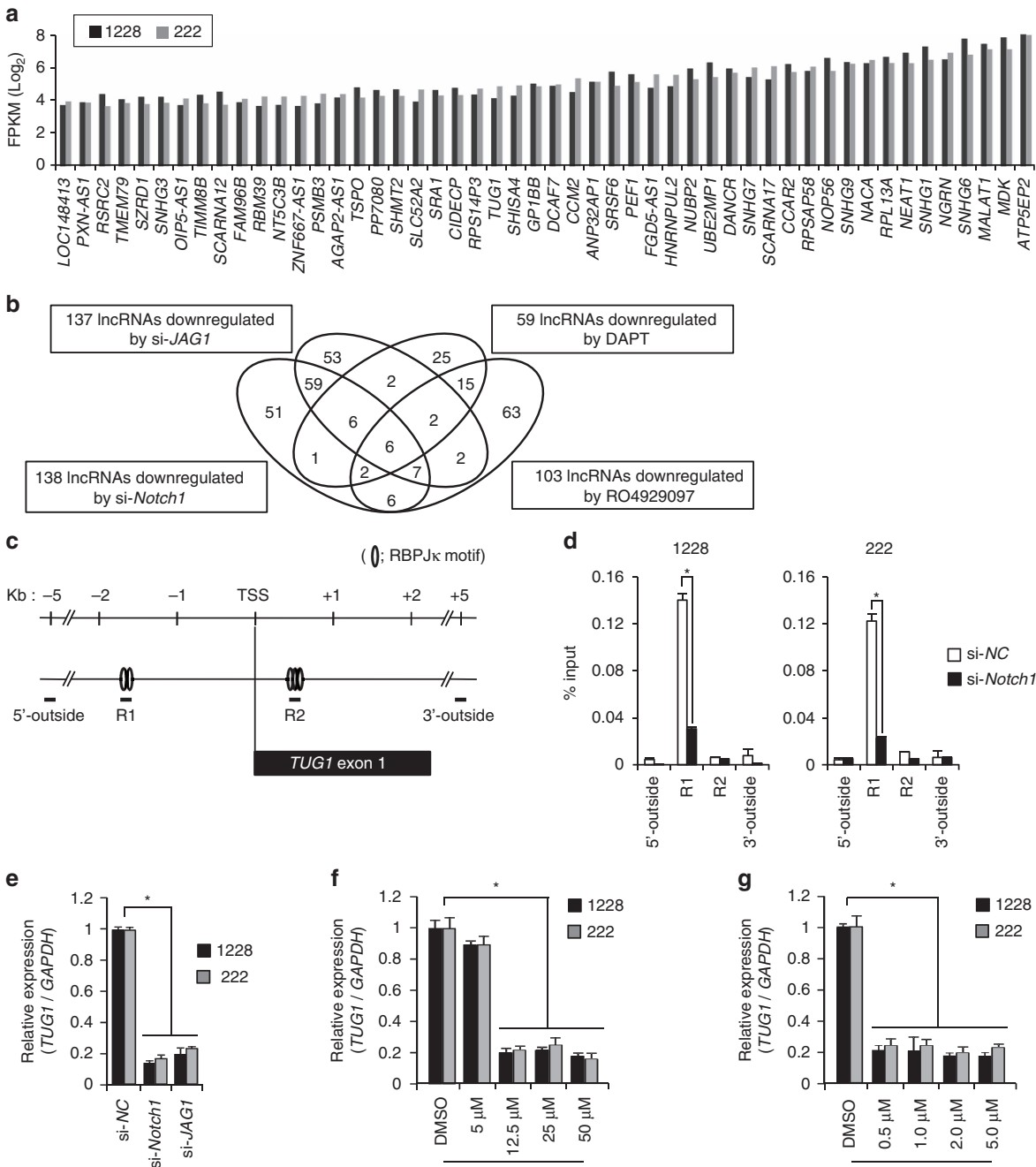

**Figure 1 | Identification of *TUG1* as a Notch downstream target in GSCs.** (**a**) Top 50 commonly highly expressed lncRNAs with Notch/RBPJκ motifs in GSCs (1228 and 222) are shown. *y*-axis indicates expression levels of each lncRNA by FPKM value (log₂). (**b**) lncRNA expression analysis of GSCs by microarray technology (1228 and 222) following use of either si-*JAG1*, si-*Notch1* or Notch signal inhibitor (γ-secretase inhibitors, DAPT and RO4929097) treatment. Venn diagram depicts the numbers of downregulated lncRNAs identified by each treatment. (**c,d**) ChIP analysis of Notch1 in the upstream region of the *TUG1* TSS. (**c**) Schematic diagram showing RBPJκ motifs around the TSS of *TUG1*. Open circles indicate RBPJκ motifs. (**d**) Enrichment of Notch1 in GSCs treated with either si-*NC* or si-*Notch1*. Regions examined by ChIP analysis are indicated as 5′-outside, R1, R2 and 3′-outside in **c**. Enrichment of Notch1 is expressed as a percentage of input DNA. *$P < 0.01$. Student's *t*-test. (**e**) Expression level of *TUG1* in GSCs treated with siRNA against the indicated genes in the *x*-axis. Relative expression level to siRNA-negative control (si-*NC*) is indicated in the *y*-axis. *$P < 0.01$. Kruskal–Wallis analysis. (**f,g**) Expression level of *TUG1* in GSCs treated with DAPT (**f**) or RO4929097 (**g**). Values are indicated relative to abundance in DMSO-treated cells. *$P < 0.01$. Kruskal–Wallis analysis. For all the experimental data, error bars indicate s.d. ($n = 3$).

cytoplasm, where these lncRNAs bind to miRNAs and quench their activity[12]. As shown in previous reports, RNA-fluorescence *in situ* hybridization (FISH) analysis revealed that *TUG1* localized to both the nucleus and the cytoplasm in GSCs (Supplementary Fig. 5a,b)[13]. We examined miRNA expression changes upon *TUG1* inhibition in two GSC lines using a miRNA-based

expression microarray approach. Twenty-one and 144 miRNAs were commonly upregulated and downregulated with a greater than twofold difference upon *TUG1* inhibition, respectively (Supplementary Data 5). Among the upregulated miRNAs, miR-145 was most prominently increased by *TUG1* inhibition (Fig. 3a–d). Intriguingly, the interaction between *TUG1* and miR-

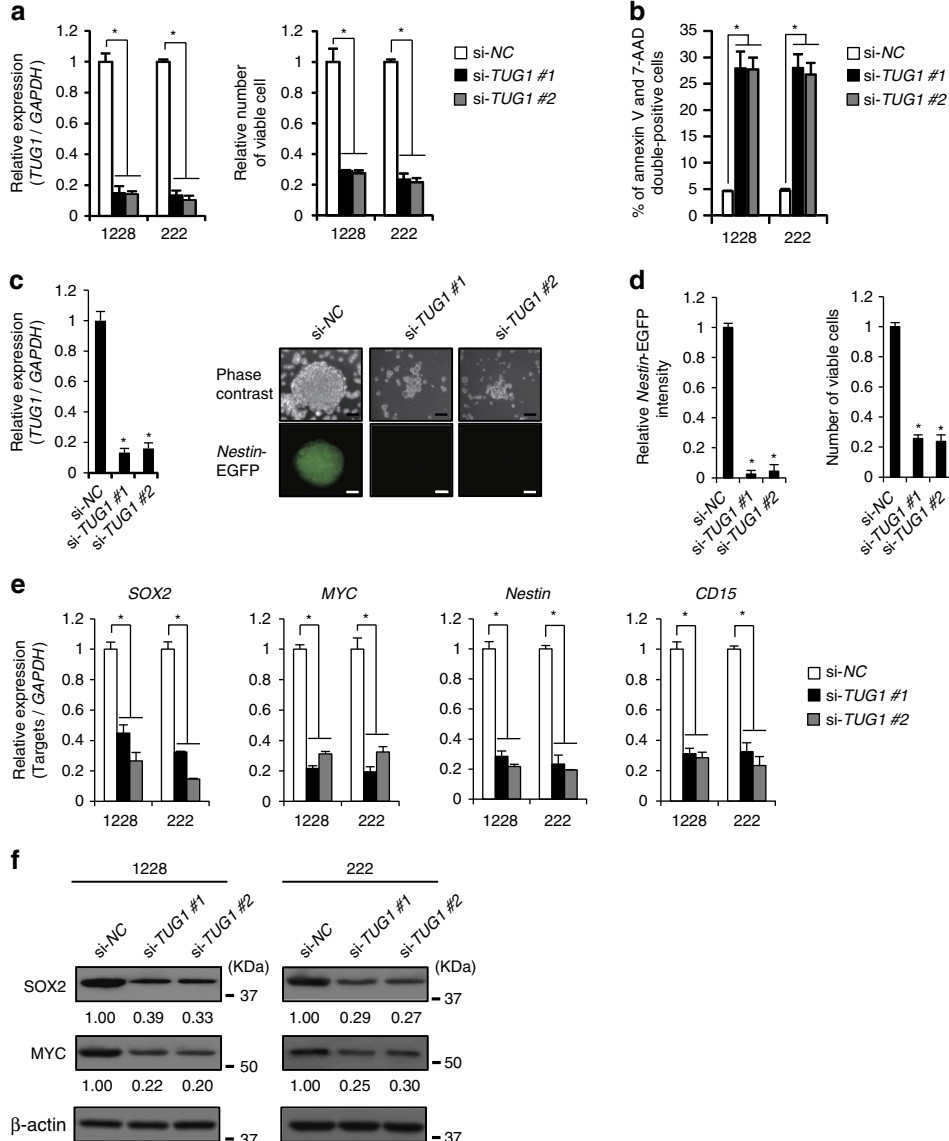

**Figure 2 | *TUG1* maintains stemness features of GSCs.** (**a**) Expression level of *TUG1* (left) and viable cell numbers (right) among GSCs treated with either si-*NC* or si-*TUG1* (si-*TUG1* #1 and #2). Viable cells were assessed by trypan blue staining. Values are indicated relative to abundance in si-*NC*-treated cells. *$P < 0.01$, Student's *t*-test. (**b**) The number of apoptotic cells in GSCs treated with si-*NC* or si-*TUG1* (si-*TUG1* #1 and #2) were counted by FACS analysis with 7-AAD and PE Annexin V staining. *$P < 0.01$, Student's *t*-test. (**c,d**) Effect of *TUG1* depletion on *Nestin* activity. (**c**) GSC-pE-Nes-222 was treated with either si-*NC* or si-*TUG1* (si-*TUG1* #1 and #2). Expression level of *TUG1* is indicated relative to abundance in si-*NC*-treated cells (left). Phase-contrast and *Nestin*-EGFP images. Scale bars, 100 μm (right). (**d**) Intensity of *Nestin*-EGFP (left) and number of viable cells (right) compared with si-*NC* control were quantified. Viable cells were assessed by trypan blue staining. *$P < 0.01$, Student's *t*-test. (**e**) Effect of *TUG1* depletion on expression of the stemness-associated genes (*SOX2*, *MYC*, *Nestin* and *CD15*). y-axis indicates relative expression level compared with that seen in si-*NC*-treated cells. *$P < 0.01$, Student's *t*-test. (**f**) Protein expression levels of SOX2 and MYC in GSCs treated with either si-*NC* or si-*TUG1* (si-*TUG1* #1 and #2). For all the experimental data, error bars indicate s.d. ($n = 3$).

145, which induced epithelial-to-mesenchymal transition through derepression of ZEB2, has been found in bladder cancer[16]. Consistently, RNA-FISH analysis revealed that numbers of *TUG1* molecules were extremely abundant compared with those of miR-145 molecules in GSCs, whereas they were significantly increased by *TUG1* inhibition (Fig. 3c,d, $P < 0.001$). Notably, as determined by RNA immunoprecipitation (RIP) analysis, both *TUG1* and miR-145 bound to wild-type AGO2 but did not bind to a mutant form of AGO2 devoid of the PAZ domain, the latter serving as a module for si/miRNA transfer in the RNA silencing pathway and as an anchoring site for the 3′ end of guide RNA within silencing effector complexes (Supplementary Fig. 5c,d)[16,28].

Using the TargetScan prediction algorithm (http://www.targetscan.org)[29], we found predicted binding sites for miR-145 not only in *TUG1* but also in *SOX2* and *MYC* mRNAs, which encode for well-known stemness-associated transcription factors[30]. Ectopic expression of miR-145 impaired GSC growth and stemness properties (Supplementary Fig. 5e–h and 6a). This phenotypic change was analogous to that observed after *TUG1* inhibition in GSCs (Fig. 2a,b). To further analyse the possible role of *TUG1* as a miRNA sponge, we used partial *TUG1* transcripts (1–2,132 nucleotides, corresponding to *TUG1* exon 1), which contained the predicted binding sites for miR-145 (Fig. 3e). Ectopic expression of these partial *TUG1* transcripts efficiently

impaired the repressive effect of miR-145 on *SOX2* and *MYC* expression, whereas mutated *TUG1* or seed sequence-deleted *TUG1* did not (Fig. 3e,f and Supplementary Fig. 6b). In addition, disruption of neurosphere formation and inactivation of the *Nestin* promoter by miR-145 were rescued by ectopic *TUG1* expression (Fig. 3g,h). These data indicate that *TUG1* protects the transcripts of stemness-related genes from miR-145-mediated degradation and aids in maintaining stemness properties.

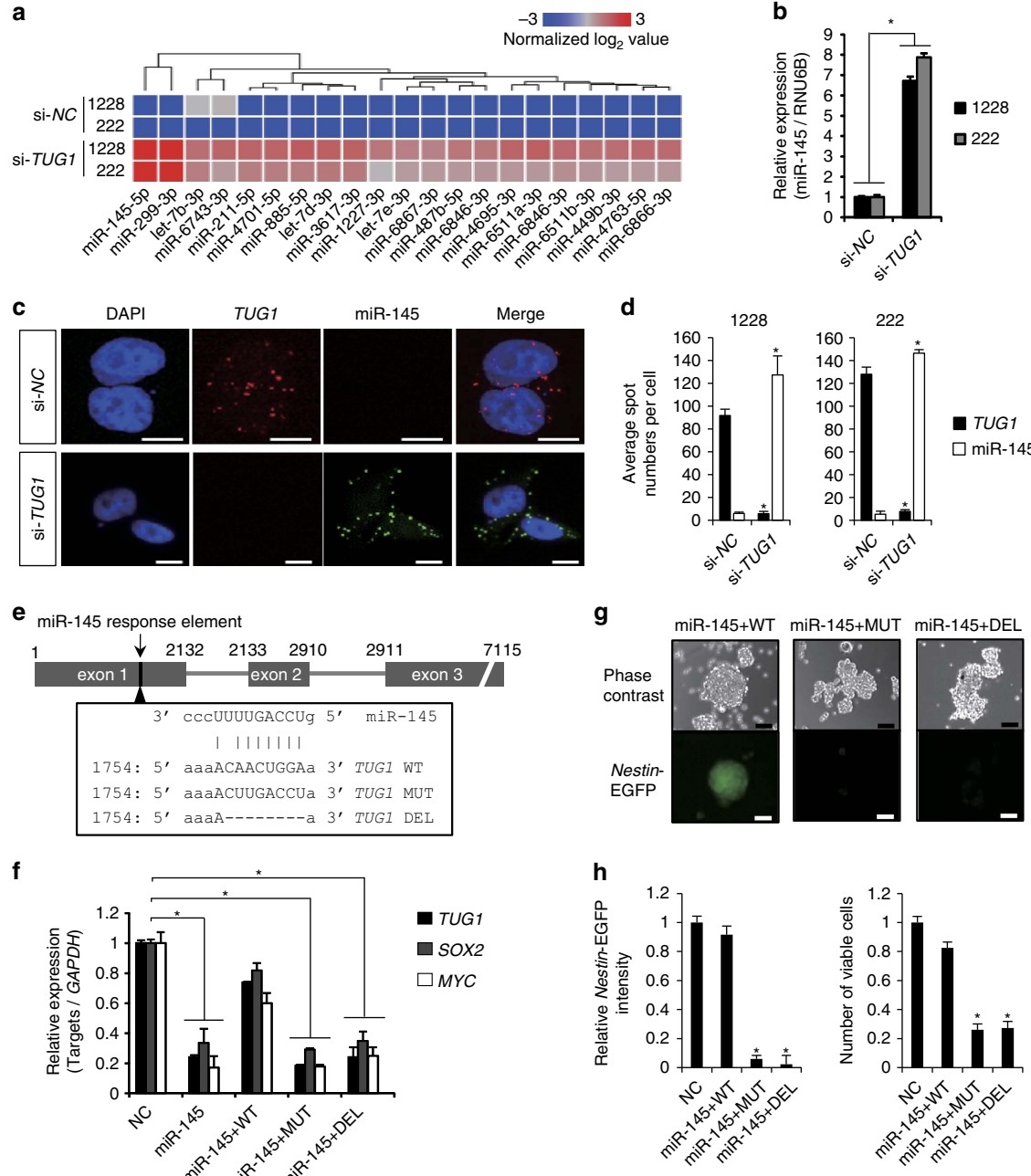

**Figure 3 | *TUG1* antagonizes miR-145 and maintains expression of stemness-associated genes.** (**a**) Heatmap shows commonly upregulated miRNAs upon inhibition of *TUG1* in GSCs (1228 and 222). Colour corresponds to expression level as indicated in the log$_2$-transformed scale bar below the matrix. Red and blue reflect high and low levels, respectively. (**b**) Effect of *TUG1* depletion on miR-145 expression in GSCs (1228 and 222). y-axis indicates relative miR-145 expression level compared with that in si-*NC*-treated cells. Expression levels were normalized to internal RNU6B. *$P < 0.01$, Student's t-test. (**c**) RNA-FISH analysis of *TUG1* (red) and miR-145 (green) in GSCs treated with si-*NC* or si-*TUG1*. Nuclei are stained with DAPI. Scale bars, 10 μm. (**d**) The spot numbers relating to *TUG1* and miR-145 detection were quantified per cell in GSCs treated with either si-*NC* or si-*TUG1*. *$P < 0.001$, Student's t-test. (**e**) Schematic of mutated (*TUG1* MUT) or deleted (*TUG1* DEL) *TUG1*. (**f**) Effect of exogenous miR-145 on *TUG1*, *SOX2* and *MYC* expression in GSC-pE-Nes-222. Partial *TUG1* transcripts (*TUG1* WT, *TUG1* MUT and *TUG1* DEL as shown in (**e**) were added to GSC-pE-Nes-222 treated with the precursor molecule of miR-145 (miR-145). Expression levels of endogenous *TUG1*, *SOX2* and *MYC* were measured. Values are indicated relative to abundance of negative control miRNA precursor (NC) treated cells. *$P < 0.01$, Kruskal–Wallis analysis. (**g**) Phase-contrast and *Nestin*-EGFP images of miR-145 + *TUG1* WT, + *TUG1* MUT or + *TUG1* DEL cells as shown in **f**. Scale bars, 100 μm. (**h**) Intensity of *Nestin*-EGFP (left) and number of viable cells (right) were quantified. Viable cells were assessed by trypan blue staining. *$P < 0.01$, Kruskal–Wallis test. For all the experimental data, error bars indicate s.d. ($n = 3$).

**TUG1-PRC2 complex suppresses neuronal differentiation genes.** Studies have demonstrated that *TUG1* in the nucleus interacts with PRC components such as Pc2 and EZH2 via its exon 2 and represses target gene expression in *trans*[13,14]. To clarify the functional roles of nuclear-localized *TUG1* in GSCs, we first transfected *TUG1* RNA labelled with 5′-bromo-uridine (BrU) (2,133– 2,910 nucleotides, corresponding to the *TUG1* exon 2) into GSCs. Immunoprecipitation analysis using an anti-BrU antibody revealed that *TUG1* bound to PRC2 components

(EZH2, SUZ12) and the YY1 transcription factor (Fig. 4a). Intriguingly, interaction between PRC2 and YY1 was disrupted by use of siRNA against *TUG1* or RNase treatment, suggesting that *TUG1* functions as a scaffold molecule between PRC2 and YY1 (Fig. 4b,c). We further determined the region within *TUG1* that interacts with YY1 and EZH2 using a series of *TUG1* deletion mutants. EZH2 and YY1 bound *TUG1* at the regions of 2,316–2,555 and 2,746–2,910 nucleotides, respectively (Fig. 5a,b). Intriguingly, these two regions are well conserved

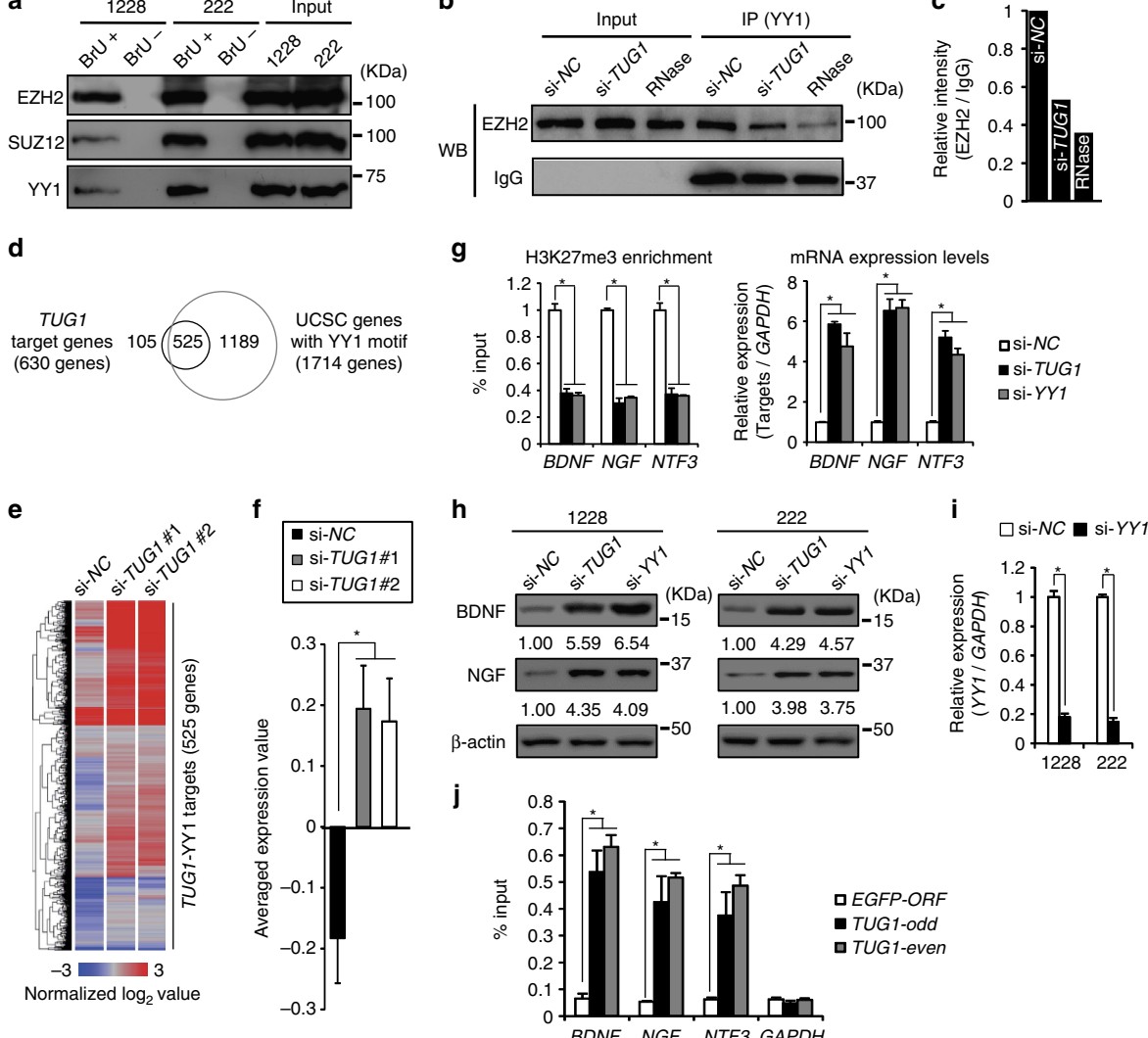

**Figure 4 | Suppression of neuronal differentiation-associated genes by the TUG1-PRC2 complex.** (**a**) RNA pull-down analysis with anti-BrU antibody. Nuclear extracts from GSCs were incubated with *TUG1* RNA labelled with BrU (BrU + ) or without (BrU-). *TUG1*-RNA-binding proteins were analysed by western blotting. Input extract is used as control. (**b**) Nuclear extracts from GSCs treated with either si-*NC* or si-*TUG1* were immunoprecipitated with anti-YY1 antibody. The immunoprecipitated fractions and lysate aliquots (Input) were subjected to western blotting. A part of the nuclear extract from si-*NC*-treated cells was also treated with RNase, immunoprecipitated with anti-YY1 antibody, and analysed by western blotting (RNase, left). Relative intensity values of EZH2 normalized to Input are indicated in (**c**). (**d**) Venn diagram shows relationship between *TUG1* target genes (630 genes) and 1,714 genes that contain a YY1 motif around their TSS. Among 630 *TUG1* target genes, 525 genes contain a YY1 motif. (**e**) Heatmap indicates expression changes of 525 *TUG1* target genes containing a YY1 motif (shown in **d**) upon inhibition of *TUG1* (si-*TUG1* #1 and #2). Colour corresponds to expression level as indicated in the log$_2$-transformed scale bar below the matrix. Red and blue reflect high and low levels, respectively. (**f**) Averaged expression value of 525 genes shown in (**e**). Error bars indicate s.e.m. *$P < 0.001$, Student's *t*-test. (**g**) Enrichment of K27me3 in the *BDNF*, *NGF* and *NTF3* promoter regions (left), and their mRNA expression levels (right) in GSCs treated with either si-*NC*, si-*TUG1* or si-*YY1*. K27me3 enrichment is expressed as a percentage of input DNA (left). Relative expression level to si-*NC* is shown (right). Error bars indicate s.d. ($n = 3$). *$P < 0.01$. Kruskal–Wallis analysis. (**h**) Protein expression levels of BDNF and NGF in GSCs treated with either si-*NC*, si-*TUG1* or si-*YY1*. (**i**) Expression changes of *YY1* in GSCs treated with si-*YY1*. Values are indicated relative to abundance in si-*NC*-treated cells. Error bars indicate s.d. ($n = 3$)*$P < 0.01$, Student's *t*-test. (**j**) ChIRP analysis for the *BDNF*, *NGF* and *NTF3* promoter regions. Probes targeting *EGFP-ORF* were used as a negative control. *GAPDH* served as a non-*TUG1* target gene control. Enrichment of *TUG1* expressed as a percentage of input DNA. Error bars indicate s.d. ($n = 3$) *$P < 0.01$, Kruskal–Wallis analysis.

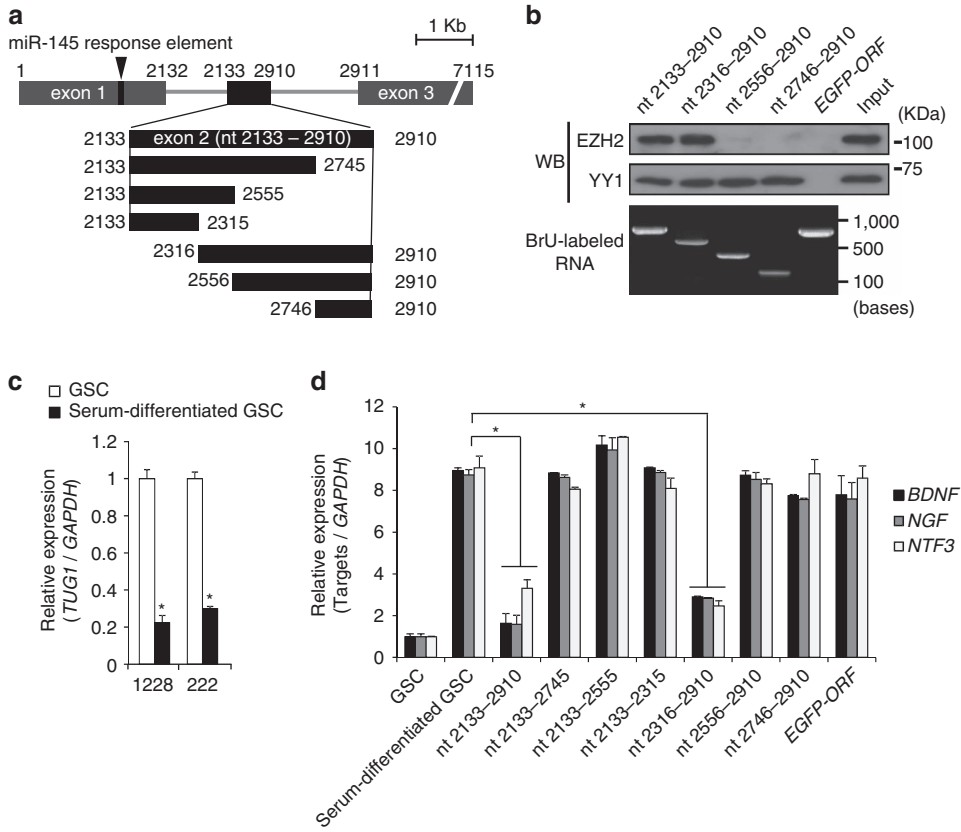

**Figure 5 | Analysis of binding region of YY1 and EZH2 in *TUG1* RNA. (a)** Schematic diagram of deletion fragments of *TUG1* exon 2 RNA. The binding site of miR-145 is indicated (arrowhead). (**b**) RNA pull-down assay with BrU-labelled partial *TUG1* RNAs as indicated in **a**. Immunoprecipitated fractions with anti-BrU antibody were analysed by western blotting using anti-EZH2 and anti-YY1 antibodies (upper). *EGFP-ORF* RNA is used as a negative control. Gel electrophoresis image of BrU-labelled RNAs is shown (bottom). (**c**) Expression level of *TUG1* in GSCs and serum-differentiated GSCs. Relative expression level compared to that in GSCs is indicated on the *y*-axis. *$P < 0.01$, Student's *t*-test. (**d**) Serum-differentiated GSCs were transfected with *TUG1* exon 2 RNA or deletion fragments as indicated in (**a**). *y*-axis indicates relative expression level of *BDNF*, *NGF* and *NTF3*. *$P < 0.01$, Kruskal–Wallis analysis. For all the experimental data, error bars indicate s.d. ($n = 3$).

between human and mouse (86.7% and 82.4% homologies, respectively).

Next, we performed a modified RNA pull-down assay coupled with promoter-microarray analysis using BrU-labelled *TUG1*-transfected cells. From this assessment, 3,183 and 2,333 genes were enriched with ectopic *TUG1* in 1228- and 222-GSCs, respectively, and 630 genes were commonly identified as targets of *TUG1* (Supplementary Data 6). Comparison between the *TUG1* target genes (630 genes) and predicted YY1 target genes (1,714 genes) determined by Whitfield's criterion[31] revealed that 525 (83.3%) contained an YY1 motif (Fig. 4d). Gene expression microarray analysis revealed that the majority of those target genes (345/525 genes, 65.7%) were upregulated upon *TUG1* inhibition (Fig. 4e,f, $P = 7.553 \times 10^{-6}$). Those *TUG1* target genes contained important regulators of neuronal differentiation, such as *BDNF*, *NGF* and *NTF3*. These target genes were silenced and modified via H3K27me3. Inhibition of *TUG1* and *YY1* derepressed their gene expression along with a reduction in the K27me3 level (Fig. 4g-i). This direct binding of endogenous *TUG1* in the *BDNF*, *NGF* and *NTF3* promoters was validated via chromatin isolation by RNA purification (ChIRP) analysis (Fig. 4j).

These findings were further supported by rescue experiments using ectopic expression of *TUG1* and its deletion mutants (Fig. 5a). Notably, endogenous *TUG1* was repressed and *TUG1* target genes (*BDNF*, *NGF* and *NTF3*) were highly expressed in serum-differentiated GSC compared with GSCs (Fig. 5c,d). Transfection of *TUG1* deletion fragments into

serum-differentiated GSC revealed that exon 2 transcript (2,133–2,910 nucleotides) and the *TUG1* fragments with EZH2- and YY1-binding domains (2,316–2,910 nucleotides) could efficiently repress the expression of *TUG1* target genes, whereas other deletion fragments could not (Fig. 5d). Taken together, these data suggest that *TUG1* physically interacted with PRC2 and promoted locus-specific K27me3 via YY1 binding activity, which resulted in suppression of neuronal differentiation-associated genes.

**Pivotal roles of the *TUG1* exon 1 region in maintenance of GSC.** To examine what are the relative and contextual contributions of miR-145 and PRC2 to the *TUG1* effects on stemness features, we overexpressed each exon of *TUG1* in GSCs treated with gamma-secretase inhibitor (RO4929097) (Fig. 5a, Fig. 6a–c). Interestingly, the *TUG1* exon 1 transcript (1–2,132 nucleotides) rescued cell growth defects, apoptosis and the expression of stemness-associated genes, whereas neither the *TUG1* exon 2 transcript (2,133–2,910 nucleotides) nor exon 3 transcript (2,911–7,115 nucleotides) rescued those features. In addition, *TUG1* exon 1 transcript efficiently rescued *Nestin* promoter activity together with impaired neurosphere formation in GSC-pE-Nes (Fig. 6d,e). Consistently, the effects of *Notch1 and JAG1* inhibition by siRNA on the stemness features of GSCs were rescued by ectopic *TUG1* exon 1 expression, but were not rescued by either exon 2 or exon 3 (Supplementary Fig. 7). Taken together, these data indicate the pivotal roles of the *TUG1* exon 1 region in maintenance of stemness features in GSCs.

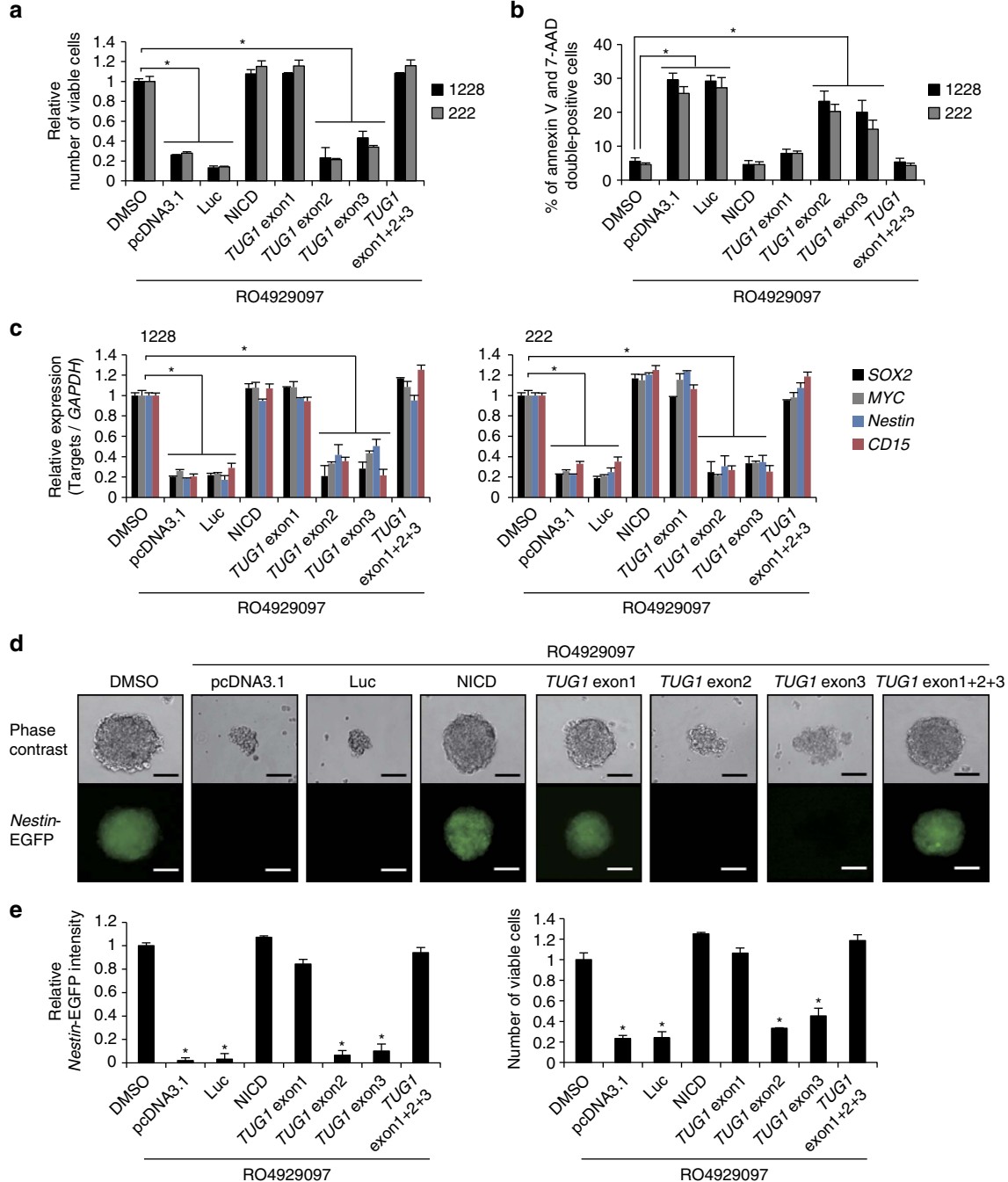

**Figure 6 | Analysis of *TUG1* transcripts for maintenance of stemness features of GSCs. (a–c)** Effects of *TUG1* overexpression on cell viability **(a)**, apoptosis **(b)** and expression of the stemness-associated genes (*SOX2*, *MYC*, *Nestin* and *CD15*) **(c)** in GSCs treated with γ-secretase inhibitor (RO4929097). Plasmid vectors expressing each *TUG1* exon (1–2,132, 2,133–2,910 and 2,911–7,115 nucleotides corresponding to exon 1, exon 2 and exon 3, respectively) were transfected. Viable cells were assessed by trypan blue staining **(a)**. The number of apoptotic cells were counted by FACS analysis with 7-AAD and PE Annexin V staining **(b)**. Expression levels of stemness-associated genes were analysed by qRT-PCR. *y*-axis indicates relative expression level compared to that seen in DMSO-treated cells **(c)**. Values are indicated relative to abundance in DMSO-treated cells. *$P < 0.01$, Kruskal–Wallis analysis. Empty vector (pcDNA3.1) and plasmid vectors expressing luciferase (Luc) were used for negative controls, whereas NICD overexpression (NICD) was used for a positive control for these experiments. **(d,e)** Effect of *TUG1* overexpression on *Nestin* activity. Plasmid vectors expressing indicated genes were added to GSC-pE-Nes-222 treated with RO4929097. Phase-contrast and *Nestin*-EGFP images were shown in **(d)**. Scale bars, 100 μm. **(e)** Intensity of *Nestin*-EGFP (left) and number of viable cells (right) compared with the DMSO control were quantified. Viable cells were assessed by trypan blue staining. *$P < 0.01$, Kruskal–Wallis analysis. For all the experimental data, error bars indicate + s.d. ($n = 3$).

**Expression analysis of *TUG1* in human GBM tissues.** To validate our findings in GSC analysis, we examined the interaction between *Notch1* and *TUG1* in clinical GBM samples. *TUG1* expression in GBM tissues ($n = 24$) was significantly higher than in normal brain tissues (Fig. 7a, $P = 6.419 \times 10^{-4}$). RNA–FISH analysis in GBM samples showed that *Notch1*-positive cells were highly enriched around perivascular regions where JAG1-positive endothelial cells may have acted as niche cells to promote GSC

self-renewal, as shown in previous studies[25]. In addition, $\sim 70\%$ of *Notch1*-positive GBM cells showed high expression of *TUG1* in the perivascular regions, whereas in areas distant from tumour vessels, the majority of GBM cells expressed neither *Notch1* nor *TUG1* (Fig. 7b–e). The positive correlation between *TUG1* and *Notch1* expression was further validated in an independent sample set using The Cancer Genome Atlas (TCGA) (Supplementary Fig. 8a).

**TUG1 promotes stemness and tumorigenicity in GSCs *in vivo*.** Finally, we investigated the molecular effect of *TUG1* inhibition on an intracranial xenograft mouse model. GSCs (GSC-pE-Nes) were inoculated into the brain of NOD-SCID mice. After 30 days of inoculation (Supplementary Fig. 9a,b), ASO-targeting *TUG1*

coupled with a potent DDS using cyclic Arg–Gly–Asp (cRGD) peptide-conjugated polymeric micelle[20], which we call hereafter *TUG1*-DDS, was administered intravenously twice a week for 4 weeks (Fig. 7f). *TUG1*-DDS was specifically accumulated and retained at least 24 h in brain tumours (Supplementary Fig. 9c,d). Treatment with *TUG1*-DDS markedly reduced tumour growth compared with control-DDS (*CTRL*-DDS) together with the downregulation of *TUG1* expression (Fig. 7g, $P = 5.451 \times 10^{-5}$). The anti tumour effect and inhibitory activity against *TUG1* expression mediated by *TUG1*-DDS were even stronger than those observed with a γ-secretase inhibitor (RO4929097, oral administration) in our model (Fig. 7g,h, $P = 9.237 \times 10^{-3}$).

RNA-FISH analysis showed that neither *TUG1* nor *Notch1*-positive cells were observed in residual tumours in mice with *TUG1*-DDS treatment, whereas in mice with *CTRL*-DDS

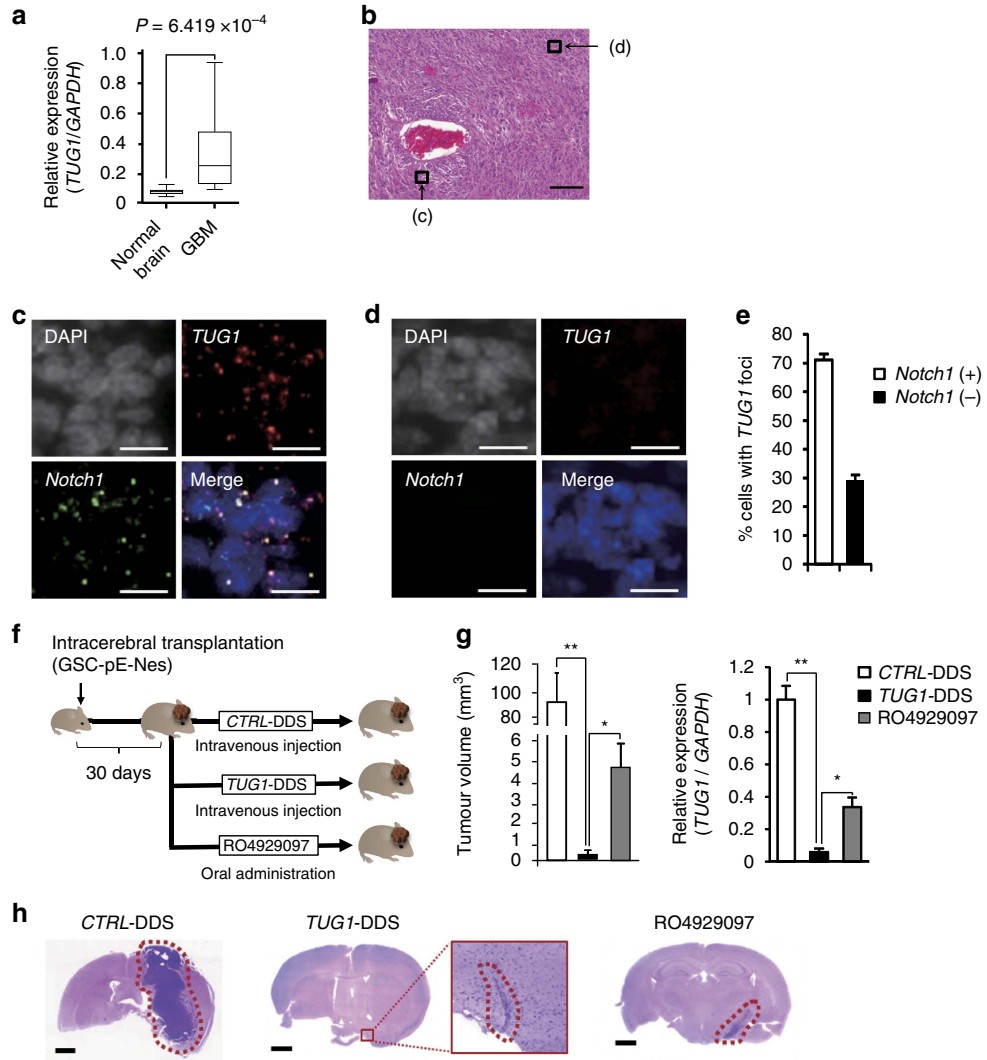

**Figure 7 | *TUG1* is a critical mediator of stemness and tumorigenicity in GSC *in vivo*.** (**a**) Expression level of *TUG1* in normal brain and GBM tissues ($n = 24$). The line inside the box represents the median, and the bottom and the top of the box are the first and the third quartiles, respectively. The whiskers indicate the range of the data. $P < 0.001$, Student's $t$-test. (**b–e**) Representation of RNA-FISH analysis of *TUG1* and *Notch1* in GBM tissues. (**b**) Hematoxylin and eosin (HE) staining of GBM section. Scale bars, 100 μm. (**c,d**) RNA-FISH analysis of *TUG1* and *Notch1* in tumour cells around area **c** (perivascular region) and area **d** (distant from blood vessel) in panel **b**. Nuclei are stained with DAPI. Scale bars, 10 μm. (**e**) Frequency of *TUG1*-positive cells in GBM specimens. Cell counts from multiple regions were averaged. Error bars indicate s.d. (**f**) GSC-pE-Nes-222 was transplanted intracranially in NOD/SCID mice. After 30 days of transplantation, *CTRL*-DDS or *TUG1*-DDS were intravenous injected twice a week for 4 weeks. RO4929097 was administered orally five times a week for 4 weeks. (**g**) Quantification of tumour volume at 4 weeks post treatment (left) and RNA expression levels of *TUG1* in the tumour cells of mouse xenograft (right). Relative expression levels to that in *CTRL*-DDS-treated tumour are indicated on the $y$-axis ($n = 4$). Error bars indicate s.e.m. *$P < 0.01$, **$P < 0.001$, Kruskal–Wallis analysis. (**h**) Representative HE-stained whole brain sections at 4 weeks post treatment. Tumour areas are surrounded by red dotted line. Scale bars, 1 mm.

treatment, *TUG1* and *Notch1* expression was predominantly observed in the perivascular regions (Fig. 8a). Consistently, *Nestin*-EGFP and CD15-positive cells were diminished in *TUG1*-DDS-treated xenografts, suggesting that inhibition of *TUG1* effectively eliminated the GSC population *in vivo* (Fig. 8b). Furthermore, in *TUG1*-DDS-treated xenografts, miR-145 was significantly increased and *MYC* and *SOX2* were decreased (Fig. 8c, $P = 2.895 \times 10^{-3}$), as was found in GSCs

*in vitro* following siRNA treatment against *TUG1* (Fig. 3c,d). Neuronal differentiation-associated genes (*BDNF*, *NGF* and *NTF3*) were reactivated in response to *TUG1*-DDS treatment (Fig. 8c,d). Taken together, these data indicate that both nuclear and cytoplasmic *TUG1* cooperatively promote the stemness and tumorigenicity of GSCs *in vivo*.

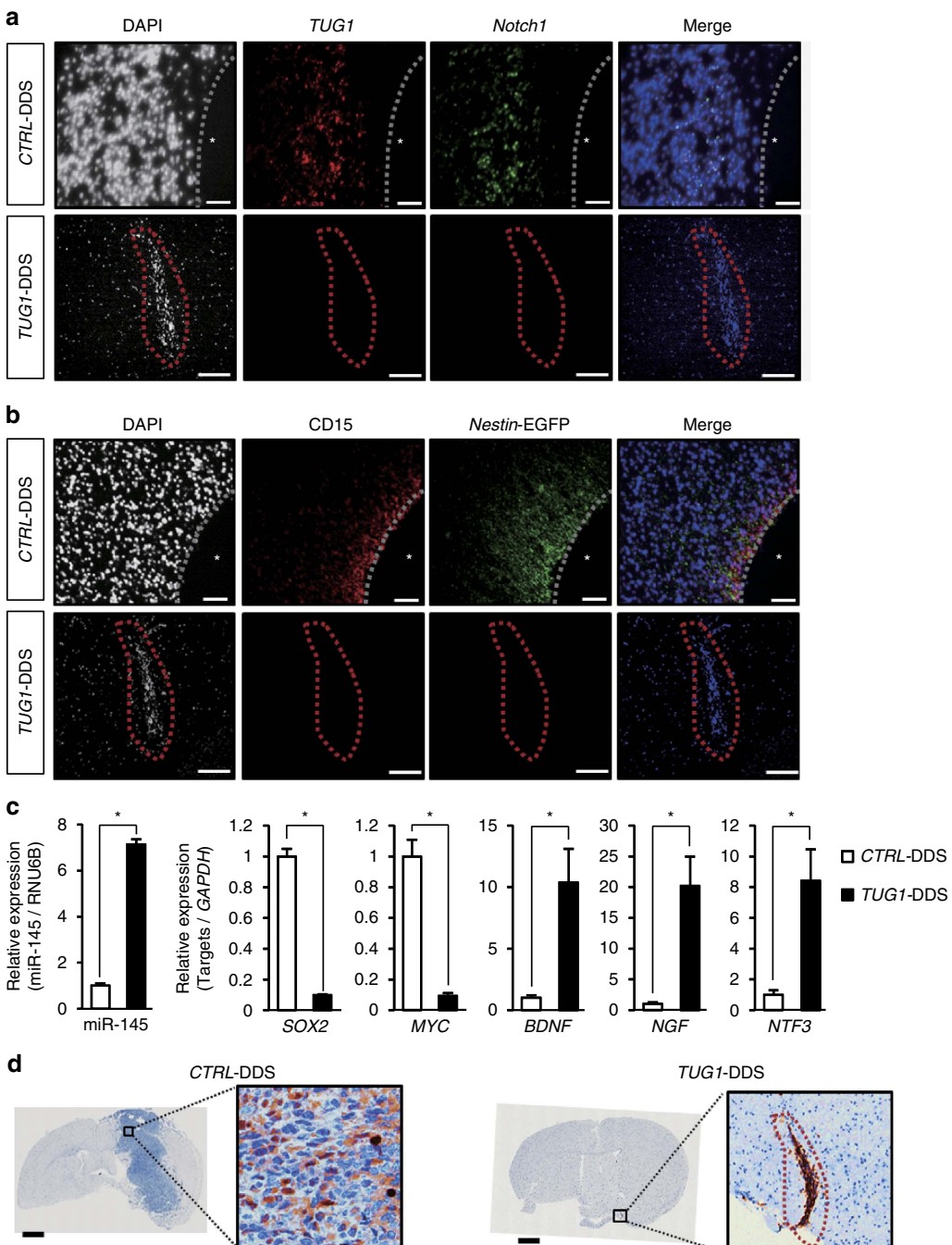

**Figure 8 | Molecular effects of *TUG1* inhibition in mouse xenograft model.** (**a**) RNA-FISH analysis of *TUG1* (red) and *Notch1* (green) in tumour cells of *CTRL*-DDS (upper panels) and *TUG1*-DDS-treated mice (bottom panels). (**b**) Immunostaining of CD15 (red) and *Nestin*-EGFP (green) in *CTRL*-DDS (upper panels) and *TUG1*-DDS-treated tumour (bottom panels). Asterisk indicates blood vessel in *CTRL*-DDS treated mice. Tumour areas are surrounded with the dashed line in *TUG1*-DDS-treated mice. Scale bars, 100 μm. (**c**) Expression levels of miR-145, *SOX2*, *MYC* and *TUG1* target genes (*BDNF*, *NGF* and *NTF3*) were examined by qPCR in tumour cells derived from the mouse xenograft. Relative expression levels compared with that in the *CTRL*-DDS-treated tumour are indicated on the y-axis ($n = 4$). Error bars indicate s.e.m. *$P < 0.001$, Student's t-test. (**d**) Immunostaining of NGF in *CTRL*-DDS (left) or *TUG1*-DDS-treated tumour (right) at 4 weeks post treatment. Tumour areas are surrounded with the dashed line in *TUG1*-DDS-treated mice. Scale bars, 1 mm.

## Discussion

An increasing number of studies have focused on the impact of Notch signalling in GSCs, which may have plasticity and respond to signals from their microenvironment[8,32]. Because GSCs contribute to the growth, survival, invasion and recurrence of brain tumours, understanding the mechanisms that govern the specific regulation of a certain set of genes, which contribute to stemness properties, by the Notch signal pathway has been actively investigated[33]. In the current study, we identified *TUG1*, which was found via an unbiased approach to be commonly upregulated in two independent GSC populations and downregulated following inhibition of Notch signalling. *TUG1* promotes self-renewal by sponging miR-145 in the cytoplasm and recruiting polycomb via YY1-binding activity to repress differentiation genes in the nucleus. We also found that *MALAT1* and *NEAT1* were highly expressed in GSCs. Although these are well known important functional lncRNAs associated with glioma behaviour[14,34,35], they did not appear to be involved in the Notch-induced GSC maintenance system (Supplementary Fig. 4f–h).

Some lncRNAs contain miRNA-binding sites and may act as miRNA sponges, such as the abundant, cytoplasmic ncRNA[36]. *TUG1* modulates miR-145 levels through its function as a miRNA sponge to trap miR-145, which is known to regulate the expression of core stemness-associated factors[37]. The miRNA sponge function of *TUG1* is supported by three lines of evidence. First, we found that cytoplasmic *TUG1* expression is extremely abundant compared with miR-145 in GSCs, whereas in *TUG1* depleted GSCs, the level of miR-145 was found to be significantly increased. Second, downregulation of the stemness-associated factors by miR-145 was rescued by exogenous *TUG1* expression, whereas *TUG1* mutants that lacked specific miR-145 seed sequences could not. Third, both *TUG1* and miR-145 were bound to wild-type AGO2 but not bound to a mutant form of AGO2 without the PAZ domain[28]. This interaction between *TUG1* and miR-145 is consistent with a previous elegant study showing that abundant and cytoplasmic *linc-RoR* functions as an endogenous miR-145 sponge, which negatively regulates mature miRNA expression levels through a posttranscriptional mechanism and avoids miR-145 increases in self-renewing hESCs[30].

Notably, *TUG1* could prevent MYC mRNA from degradation by quenching miR-145. As MYC is an important downstream factor in Notch signalling[38], as well as a transcriptional activator of *TUG1* (Supplementary Fig. 10), *TUG1* and MYC may form a reinforcing activation loop leading to mutual stabilisation of their gene expression. In any case, the Jagged1-Notch1-*TUG1* pathway can effectively enhance GSC self-renewal by antagonising miR-145 activity. Intriguingly, in addition to targeting stemness-associated factors, miR-145 is also known to act as a tumour suppressor by decreasing cell growth, apoptosis and angiogenesis in various cell types including those derived from GBM[16,39]. These tumour-suppressive functions of miR-145 may also explain why inhibition of *TUG1* efficiently induced apoptosis after partial neuronal differentiation.

In the nucleus, *TUG1* selectively regulated the epigenetic status of neuronal differentiation-associated genes, such as *BDNF*, *NGF* and *NTF3*. Notably, BDNF acts as a tumour suppressor in gliomagenesis[40]. Theoretically, RNA such as *TUG1* may freely diffuse and transmigrate between chromosomes, but some DNA elements may be required for efficient loading onto certain genomic loci[41]. In this study, we illustrated that YY1 functional bridges facilitated the loading of regulatory *TUG1* to its DNA targets. It is possible that YY1 functional bridges alone may not be sufficient to specify the target loci, as *TUG1* was not simultaneously enriched at a large number of other YY1 sites.

However, we found that *TUG1* fragments with EZH2- and YY1-binding domains could efficiently repress the expression of the PRC2-*TUG1*-YY1 target genes, whereas other deletion fragments, which lack either EZH2 or YY1 binding domains (dominant-negative constructs), could not. Therefore, *TUG1* in the nucleus appeared to have interacted with PRC2 and promoted locus-specific methylation of histone H3K27, at least in part, via YY1-binding activity. Of interest, both the regions where PRC2 and YY1 interact with *TUG1* are evolutionarily conserved between mouse and human, suggesting the importance of these regions[42]. Consistent with our findings, the concept of YY1 as docking protein has also been illustrated in an X chromosome inactivation model, in which YY1 tethers *Xist* RNA-PRC2 to the inactive X nucleation center[41].

These regional distinct functions of *TUG1* may raise the question of what are the relative and contextual contributions of each region via miR-145 and PRC2 on stemness features. Importantly, attenuation of the Notch-*TUG1* axis by siRNA or γ-secretase in regulating self-renewal of glioma cells was efficiently rescued by *TUG1* exon 1 transcript where the miR-145-binding site locates indicating that there is a pivotal role for the exon 1 and miR-145 interaction in maintenance of stemness properties.

Many strategies to improve the delivery of drugs to GBMs are under investigation, as the blood–brain barrier and the blood–brain tumour barrier limit the enhanced permeability and retention effect, thereby resulting in poor efficacy due to inefficient drug delivery to GBM[43]. Brain tumour vessels show overexpression of receptors that mediate ligand-dependent drug delivery[44]. In particular, RGD peptides are promising ligand molecules for targeting $\alpha_v\beta3$ and $\alpha_v\beta5$ integrins, which are frequently overexpressed in GBM cells[45]. In order to inhibit *TUG1* expression in a mouse xenograft model, we used cRGD ligand-conjugated polymeric micelles for delivery. These targetable polymeric micelles retained ASO accumulation within tumours, which is associated with transcytosis-mediated penetration and enhanced gene silencing activity. Although further investigations are required, cRGD-mediated drug delivery is a powerful strategy for targeting GBMs through facilitated ASO delivery beyond the blood–brain tumour barrier.

Emerging insights into gliomagenesis have revealed that GSCs have the potential to initiate and maintain the growth of gliomas, and may be crucial for their resistance to conventional therapies[1,46]. Further, the existence of GSCs and plastic epigenetic regulation may be linked to the morphological and lineage heterogeneity that is observed in GBM[2,5]. In the current study, we addressed the novel mechanism underlying the contribution of the Notch signalling pathway to the stemness features of GSCs in response to contact-dependent signals from adjacent cells expressing Jagged1. Further, we provide a new paradigm whereby targeting *TUG1*, especially coupled with a potent DDS, is an effective novel strategy for GBM treatment. One limitation of the current study is that both our GSC lines were classified as proneural type GBM by gene expression analysis (Supplementary Fig. 8b). Indeed, expression levels of *TUG1* and *Notch1* were prominently upregulated in proneural and classical GBM compared with two other subtypes (neural and mesenchymal) (Supplementary Fig. 8c). Therefore, it might be possible that targeting *TUG1* is more promising for treatment of proneural or classical GBMs rather than other subtypes. However, given that inhibition of γ-secretase-mediated Notch cleavage is a primary focus for the development of targeted therapeutics in GBM (including application in clinical trials such as RO4929097)[47], our data

may provide a strong rationale that targeting *TUG1* is a more specific and potent therapeutic approach to eliminate the GSC population.

## Methods

**GBM tissues, cell culture and drug treatment.** GBM tissue samples were obtained from patients undergoing surgical treatment at Nagoya University hospital, Japan, after they provided written informed consent. Total RNA of normal brain was purchased from BioChain (Hayward, CA, USA). The procedures used for derivation of GSCs (1228-GSC and 222-GSC) were as described previously[5]. In brief, dissociated tumour cells were cultured in Neurobasal Medium (Life Technologies, Carlsbad, CA) containing N2 and B27 supplements (Life Technologies), along with human recombinant basic fibroblast growth factor and epidermal growth factor (20 ng/ml each; R&D Systems, Minneapolis, MN, USA). Serum-differentiated GSCs were established by culturing GSCs in DMEM (Life Technologies) containing 10% fetal bovine serum[5]. Serially transplanted GSCs were generated according to previously published methods[1]. GSCs were routinely tested for mycoplasma contamination in all the experiments. For blocking of Notch signalling in GSCs, the γ-secretase inhibitor DAPT (N-[N-(3,5-difluorophenacetyl)-l-alanyl]-S-phenyl glycine t-butyl ester), Sigma-Aldrich, St Louis, MO, USA) or RO4929097 (Selleck Chemicals, Houston, TX, USA) were added, and the medium was changed every other day. For *in vivo* studies, RO4929097 was formulated as a suspension in 1% carboxymethyl cellulose with 0.2% Tween 80 for oral administration.

**RNA extraction and preparation for RNA sequencing.** Procedures utilized for next-generation sequencing were as described previously[48]. In brief, total RNA was extracted from GSCs using TRIzol (Life Technologies). The RNA-seq libraries were prepared using a paired-end RNA Sequencing Sample Prep Kit (Illumina, San Diego, CA, USA). Two microgram of total RNA was used as the starting material, and polyadenylated RNA was selected using Sera-Mag Magnetic Oligo (dT) Beads (Illumina). First-strand cDNA synthesis was performed using SuperScript II (Life Technologies), and second-strand cDNA synthesis was performed using DNA Pol I in the supplied GEX second-strand reaction buffer (Illumina). Paired-end adaptors were ligated to the cDNA fragments. PCR (eight cycles) was performed with Phusion High-Fidelity DNA Polymerase (Finnzymes Oy, Espoo, Finland). The PCR products were cleaned up with Agencourt AMPure XP magnetic beads (Beckman Coulter, Miami, FL, USA). cDNA was applied to the flow cell and paired-end 76 nucleotide-long reads were generated using Illumina GAIIx and Miseq instruments. We performed one experiment for the RNA-seq analysis in each cell line. Obtained data were aligned with TopHat 2.0.13 to hg19 with default parameters. Expression differences between two GSC lines were calculated with Cuffdiff 2.2.1.

**lncRNA expression analysis via microarray technology.** Total RNA was isolated from GSCs with TRIzol reagent (Life Technologies). RNA was amplified into cRNA and labelled according to the Agilent One-Colour Microarray-Based Gene Expression Analysis protocol (Agilent Technologies, Santa Clara, CA). Labelled samples were purified with the RNeasy Kit (Qiagen, Valencia, CA) and hybridized to SurePrint G3 Human GE 8 × 60 K array slides (G4851B, Agilent Technologies) at 65 °C with rotation at 10 rpm for 17 h. The arrays were scanned using an Agilent Microarray Scanner (G2565BA, Agilent Technologies). The scanned images were analysed using the Feature Extraction software, version 10.7.3.1 (Agilent Technologies) with background correction. Data analysis was performed with GeneSpring GX, version 12.6.0 (Silicon Genetics). Expression data were centred on a median with the use of the GeneSpring normalisation option with no substantial difference in results. The microarray analysis was performed in duplicate for each cell line. Statistical comparisons were made with the use of GeneSpring volcano plot filtering.

**RNA interference.** For downregulation of gene expression with siRNA, GSCs were transfected with 20 nM siRNA targeting each gene or control non-targeting siRNA (negative control siRNA) (AM4611, Applied Biosystems, Foster City, CA, USA) using Lipofectamine 2000 (Life Technologies). Seventy-two hours after the transfection of siRNA, total RNA was extracted from GSCs and the effects were validated. Targeted sequences for each gene are listed in Supplementary Data 4.

**Quantitative reverse transcription-PCR (qRT-PCR).** Total RNA was isolated using TRIzol (Life Technologies), and 1 μg was used for reverse transcription with the SuperScript VILO cDNA Synthesis Kit (Life Technologies). Expression levels of target genes were determined using the delta Ct method, and normalized to the housekeeping gene *GAPDH*. Target genes were measured using at least three replicates by TaqMan PCR or SYBR Green quantitative PCR (qPCR). TaqMan PCR assays (Applied Biosystems) and oligonucleotide primers are shown in Supplementary Data 4.

**Immunohistochemistry.** GSCs were fixed with freshly made 2% paraformaldehyde and immunostained with anti-SOX2 (#3579, 1:500, Cell Signaling Technology, Danvers, MA, USA), anti-MYC (#5605, 1:500, Cell Signaling Technology) and anti-GFP (M048-3, 1:100, MBL International, Nagoya, Japan) antibodies. Cryosections of mouse xenograft tumour were fixed with freshly made 4% paraformaldehyde and immunostained with anti-CD15 (ab135377, 1:100, Abcam) and anti-GFP (M048-3, 1:100, MBL International) antibodies. Primary antibody-antigen complexes were visualized using anti-mouse Alexa Fluor 488 and anti-rabbit Alexa Fluor 546 secondary antibodies (Molecular Probes, 1:500, Eugene, OR, USA). Nuclei were counterstained using 4′, 6- diamidino-2-phenylindole (DAPI). Images were obtained with a Leica DMI6000B microscope (Leica Microsystems, Wetzlar, Germany). Approximately 100 cell nuclei within at least three views were analysed. For hematoxylin–eosin staining, mouse xenograft tumours were fixed in 4% PFA for 24 h and washed in PBS. Fixed tumour tissues were embedded in paraffin, sectioned and stained with hematoxylin–eosin. Immunohistochemistry for NGF was performed on paraffin sections using a primary antibody against NGF (ab52918, 1:200, Abcam) and a horseradish peroxidase (HRP)-conjugated IgG (A18757, 1:500, Life Technologies), and the proteins *in situ* were visualized with 3, 3-diaminobenzidine.

**Chromatin immunoprecipitation (ChIP) assay.** ChIP assays were performed based on a modification of previously published methods[5]. In brief, GSCs were treated with 0.5% formaldehyde to cross-link histones to DNA followed by the addition of 125 mM glycine to stop the cross-linking reaction. After sonication of cell pellets, the lysate was incubated with 4 ul of anti-K27 trimethylated histone H3 (ref. 49) and 10 μl of either anti-Notch1 (ab27526, Abcam), anti-K27 trimethylated histone H3 (ab6002, Abcam), anti-YY1 (ab12132, Abcam) and anti-MYC (ab32, Abcam) antibodies. ChIP products were analysed by SYBR Green ChIP-qPCR using the primer sets shown in Supplementary Data 4.

**Flow cytometry.** Cells were analysed by fluorescence activated cell sorting using the PE Annexin V Apoptosis Detection Kit I (BD Bioscience, San Jose, CA, USA). In brief, single-cell suspensions were prepared from GSC spheres, and stained with PE Annexin V and 7-aminoactinomycin D. Data were collected on a FACS Calibur (BD Bioscience) and analysed using CELL Quest Pro (BD Bioscience).

**Western blot analysis.** For western blot analysis, anti-SOX2 (#3579, 1:1000, Cell Signaling Technology), anti-MYC (#5605, 1:1000, Cell Signaling Technology), anti-BDNF (ab108383, 1:1000, Abcam), anti-NGF (ab52918, 1:1000, Abcam), and anti-β-actin (#4967, 1:2000, Cell Signaling Technology) were used as the primary antibodies. HRP-linked anti-mouse IgG (#7076, 1:1000, Cell Signaling Technology) and HRP-linked anti-rabbit IgG (#7074, 1:1000, Cell Signaling Technology) antibodies were used as secondary antibodies. Uncropped scans of the blots are provided in Supplementary Fig. 11.

**Exogenous expression of *TUG1*.** *In vitro* transcription of *TUG1* RNA was performed with DNA templates that correspond to the different regions of the *TUG1* RNA sequence (1–2,132 corresponding to exon 1 full-length, 2,133–2,910 corresponding to exon 2 full-length, 2,316–2,910, 2,556–2,910, 2,746–2,910, 2,133–2,745, 2,133–2,555 and 2,133–2,315 nucleotides) using the CUGA7 *in vitro* Transcription Kit (Nippon Gene, Tokyo, Japan). The DNA templates were amplified by PCR with the primer sets listed in Supplementary Data 4. To append the 5′-cap structure and 3′-polyA tail, the DNA template was ligated into the pTnT Vector (Promega, Madison, WI, USA). The seed sequence of miR-145 (5′- AAC TGGA-3′) within *TUG1* RNA exon 1 (1–2,132 nucleotides) was mutated or deleted using the QuikChange Site-Directed Mutagenesis Kit (Stratagene, La Jolla, CA, USA) with the primer sets listed in Supplementary Data 4. In addition, PCR-generated fragments containing cDNA for *NICD* (not including the si-Notch1 target region), *TUG1* exon 1 (1–2,132 nucleotides), *TUG1* exon 2 (2,133–2,910 nucleotides), *TUG1* exon 3 (2,911–7,115 nucleotides) and *luciferase* were also ligated into a pcDNA3.1 vector (Life Technologies), which were subsequently transfected into the glioma cells in order to overexpress those *TUG1* transcripts. Primer sequences for plasmid construction are summarized in Supplementary Data 4. Obtained plasmids were verified by conventional sequencing analysis.

**miRNA microarray.** The SurePrint G3 Human miRNA 8x60K Microarray (G4872A, Agilent Technologies) was used according to the manufacturer's protocols. Total RNA was isolated from GSCs with TRIzol reagent (Life Technologies). In total, 100 ng of total RNA was labelled with pCp-Cy3 (Agilent Technologies) and 15 units of T4 RNA ligase (GE Healthcare, Little Chalfont, Buckinghamshire, UK) at 16 °C for 2 h. Labelled samples were purified with MicroBio-Spin six columns (Bio-Rad, Richmond, CA, USA) and hybridized to microarrays at 55 °C with rotation at 20 rpm for 20 h. The arrays were scanned using an Agilent Microarray Scanner (G2565BA, Agilent Technologies). The scanned images were analysed using the Feature Extraction software, version 10.7.3.1 (Agilent Technologies) with background correction. Data analysis was performed with GeneSpring GX, version 12.6.0 (Silicon Genetics). Expression data were normalized to the 75th percentile using the GeneSpring normalisation option with no substantial difference in

results. The miRNA microarray analysis was performed in duplicate for each cell line. Statistical comparisons were made with the use of both the GeneSpring analysis-of-variance tool and volcano plot filtering.

**RNA-FISH analysis.** RNA was visualized in paraffin-embedded sections using the QuantiGene ViewRNA ISH Tissue Assay Kit (Affymetrix, Frederick, MD, USA). In brief, tissue sections were rehydrated and incubated with proteinase K. Subsequently, they were incubated with ViewRNA probe sets designed against human *TUG1,* miR-145 and *Notch1* (Affymetrix).

**Induction of miR-145 in GSCs.** GSCs were transfected with a precursor molecule mimicking miR-145 (Pre-miR-145 precursor, final concentration of 30 nM, Applied Biosystems) or negative control miRNA (Pre-miR Negative Control #1, 30 nM, Applied Biosystems) according to the manufacturer's instructions. The expression level of miR-145 was measured by TaqMan MicroRNA Assays (Applied Biosystems) and normalized to RNU6B. Assay IDs used are listed in Supplementary Data 4.

**RIP analysis.** RIP analysis was performed using RiboCluster Profiler RIP-Assay Kit (RN1005, MBL International) according to the manufacturer's protocols. RNA-protein complexes were immunoprecipitated with an anti-AGO2 antibody (RN005M, MBL International), anti-Flag antibody (F1804, Sigma-Aldrich), and anti-IgG antibody as a negative control (PM035, MBL International). Immunoprecipitated RNA was then analysed by qPCR using the primers listed in Supplementary Data 4. For analysis of the interaction between RNA and the AGO2 PAZ domain, GSCs were transfected with expression vectors encoding either Flag-AGO2 or Flag-AGO2 lacking the PAZ domain. The AGO2 expression vector (#21538) was purchased from Addgene (Cambridge, MA, USA). The Flag-AGO2 variant lacking the PAZ domain was generated with the QuikChange Site-Directed Mutagenesis Kit (Stratagene) using the following primers—forward; 5′-CTGTGCC TTGTAAAACGCT-3′, and reverse; 5′-GCAGGACAAAGATGTATTAAAAAA-3′. The obtained vector was verified by conventional sequencing analysis.

**Analysis of RNA-binding proteins by RNA pull-down assay.** Nuclear extracts were obtained from GSCs ($1 \times 10^8$) and incubated with 50 pmol of BrU-labelled *TUG1* RNA. RNA and nuclear extract conjugants were mixed with an anti-BrU antibody (MI-11-3, MBL International) for 2 h at 4 °C. To collect the immuno-precipitated RNA-binding proteins, Dynabeads Protein G (Life technologies) were added and incubated for 1 h at 4 °C. These *TUG1* RNA-binding proteins were analysed by western blot analysis with anti-EZH2 antibody (#3147, Cell Signaling Technology), anti-YY1 antibody (ab12132, Abcam, Cambridge, UK), and anti-IgG antibody as a negative control (PM035, MBL International). HRP-linked anti-mouse IgG (#7076, Cell Signaling Technology) and HRP-linked anti-rabbit IgG (#7074, Cell Signaling Technology) antibodies were used as secondary antibodies.

**Analysis of *TUG1* RNA-binding loci by ChIP assay *in vivo*.** Ten nM BrU-labelled *TUG1* RNA (2,133–2,910 nucleotides) was transfected into GSCs using Lipofectamine 2000 (Life Technologies). After 72 h of transfection, GSCs were cross-linked with 0.5% formaldehyde, and chromatin fractions were extracted and homogenized. Chromatin and BrU-labelled *TUG1* RNA mixtures were immunoprecipitated using an anti-BrU antibody and Dynabeads Protein G. Purified chromatin was eluted to yield DNA, which was then subjected to microarray analysis or ChIP-qPCR analysis.

Equivalent amounts of DNA, which was extracted after immunoprecipitation for BrU-labelled *TUG1* RNA, and total input DNA were amplified in parallel using a random primer method with the GenomePlex Complete Whole Genome Amplification Kit (15 cycles), according to the manufacturer's instructions (Sigma-Aldrich). Amplified products were labelled with Cy5 for immunoprecipitated DNA samples and Cy3 for input DNA using the BioPrime Array CGH Genomic Labeling System (Life Technologies). A total of 4 µg of labelled DNA was then hybridized to the human promoter ChIP-chip microarray (G4874A, Agilent Technologies) for 48 h at 65 °C. Data were extracted from scanned images using the Feature Extraction software, version 10.7.3.1 (Agilent Technologies). The text files were then imported into Genomic Workbench, standard edition 5.0.14 (Agilent Technologies), for analysis.

**ChIRP assay.** ChIRP assays were performed as described[50]. GSCs were cross-linked with 1% glutaldehyde for 15 min at room temperature and then quenched with 125 mM glycine for 5 min. The cross-linked chromatin was then isolated and hybridized with the probes, followed by streptavidin bead-based capturing and wash/elution steps. The antisense DNA probes targeting *TUG1* and *EGFP*-ORF sequences were designed using ChIRP Probe Designer version 4.2 (https://www.biosearchtech.com/chirp-designer) and divided into odd and even pools. Probes used for ChIRP assays are listed in Supplementary Data 4. The ChIRP captured chromatin was then reverse cross-linked and analysed by qPCR using the primers listed in Supplementary Data 4.

**Analysis of TCGA data.** For analysis of gene expression (*TUG1, Notch1, SOX2, MYC, BDNF, NGF* and *NTF3*) in clinical GBM samples, we used the level-3 preprocessed expression data of Affymetrix HT Human Genome U133a microarray platform from The TCGA (https://tcga-data.nci.nih.gov/).

**Polymeric micelle preparation for administration *in vivo*.** A cyclic-Arg-Gly-Asp (cRGD) peptide-conjugated polymeric micelle[20] was used for systemic delivery of ASOs *in vivo*. Sequences used are as follows (1) Firefly GL3 luciferase (*CTRL*): 5′-TCGAAGTACTCAGCGTAAGTT-3′; (2) *TUG1*: 5′- TGAATTTCAATCATTT GAGAT -3′. For observation of ASO *in vivo*, ASOs were labelled with Alexa647.

**Animal experiments.** All experiments were performed under protocols approved by the Institutional Animal Care and Use Committee of Nagoya City University Graduate School of Medical Sciences. 222-GSC-pE-Nes were injected intracranially into 6-week-old female NOD/SCID mice ($n = 12$, SLC, Shizuoka, Japan). Four weeks after the injection of GSCs, CTRL-DDS ($n = 4$) or TUG1-DDS ($n = 4$; 1 mg/kg per day) were intravenously injected twice a week for 4 weeks. RO4929097 ($n = 4$; 20 mg/kg per day) was administered orally five times a week for 4 weeks. The accumulation of ASOs in tumour tissue was confirmed by an *in vivo* spectral imaging system (IVIS Lumina II, Xenogen, Alameda, CA, USA). Tumour volumes were calculated using the formula (width, height and length; W × H × L)/2.

**Statistics.** For the statistical analysis of clinical data sets from TCGA, we used R Statistical Software (version 2.14.0; R Foundation for Statistical Computing, Vienna, Austria). The statistical significance of the differences between two groups was analysed by paired Student's *t*-test. Kruskal–Wallis analysis were used for to evaluate the extent of differences among more than three groups (StatView software version 5.0; Abacus Concepts, Berkeley, CA, USA). Microarray data analysis were performed by paired Student's *t*-test. All reported *P* values were two-sided, with $P < 0.01$ considered statistically significant.

**Data availability.** The Gene Expression Omnibus accession numbers for the RNA-seq data, ChIP-chip microarray data, gene expression microarray, miRNA expression microarray for 1228- and 222-GSCs are GSE66200, GSE86348, GSE65910, GSE79897 and GSE79896, respectively.

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

## Acknowledgements

This research was supported by the Project for Development of Innovative Research on Cancer Therapeutics (P-DIRECT) and the Project for Cancer Research and Therapeutics Evolution (P-CREATE) from Japan Agency for Medical Research and development, AMED (Y. Kondo), by the PRESTO, JST (Y. Kondo), and by the Grant-in-Aid for Scientific Research, the Japan Society for the Promotion of Science (25290048, Y. Kondo).

## Author contributions

Conception and design: K.K., Y.K.; development of methodology: K.K., F.O., H.K., Y.K.; acquisition of data: A.N., K.S., S.S., S.T., N.I.; analysis and interpretation of data: T.S., Y.T., K.K., A. H., Y.K.; writing, review and/or revision of the manuscript: K.K., Y.K.; administrative, technical, or material support: H.K., M.N., H.J.K., K.M., K.K., Y.K.

## Additional information

**Competing financial interests:** The authors declare no competing financial interests.

