## [Peer Review File · Nature Communications]

Reviewer #1 (Remarks to the Author)

This interesting paper describes that Notch1 activation in glioma stem cells specifically induces expression of the lncRNA TUG1 which promotes cellular self-renewal through a somewhat complex signaling pathway, in which repression of differentiation genes occurs through locus-specific methylation of histone H3K27. Based on this discovery (which seems to me interesting but I'm not competent to give a relevant opinion concerning the originality and relevance of this signaling pathway as a drug target), intravenous treatment with antisense oligonucleotides targeting TUG1 coupled with drug delivery system induces glioma stem cells differentiation. This approach efficiently represses stem cells growth in vivo. In fact, the delivery system is composed of polymer micelles surface functionalized with cyclic-Arg-Gly-Asp (cRGD) peptide and loaded with oligonucleotides with the relevant silencing sequence. The nanocarrier used is not novel as disclaimed by the authors themselves (see Christie et al., ACS Nano 2012 and Miura et al., ACS Nano 2013 cited in the paper as references 26 and 27). Nevertheless, by using this delivery system, the volume of brain glioma was impressively decreased which makes this paper of interest.

Reviewer #2 (Remarks to the Author)

The study by Katsushima et al. uncovers the lnc-RNA TUG1 as a Notch-regulated transcript in glioblastoma stem cells (GSC). It shows that Notch upregulates TUG1 and that TUG1 acts partly by sponging miR-145 and partly via PRC2 to regulate genes that regulate differentiation. Inhibition of TUG1 in vivo with ASO coupled to a DDS led to inhibition of orthotopic xenograft growth. TUG1 and Notch expressions in human GBM samples were consistent with Notch regulation of TUG1 in GSCs.

The data shown in the manuscript are interesting, significant for GSC biology and partly novel. The regulation of TUG1 by Notch and the effects of Notch on GSCs are new. The interactions between Notch and miR-145 and PRC2 have been shown before in other systems but are new for GSC and GBM. The therapeutic data are particularly compelling. The work contains complementary mechanistic, expression and experimental therapeutic components and is mostly well performed. The manuscript is clearly written. However, the findings have a few shortcomings that need to be experimentally addressed. More specifically, the following issues should be addressed:

- 1- The data are almost entirely reliant on loss of function (mostly siRNA-based) approaches and should be complemented with gain of function approaches. Specifically, the authors should use Notch gain of function (e.g. NICD overexpression) to show the effects of Notch on TUG1. Also, TUG1 overexpression should be used to assess the effects of TUG1 on molecular and functional endpoints in GSCs.
- 2- The functional assays are generally sparse and not fully convincing. With one limited exception (Fig 2B), the readout is a marker (Sox2, nestin, CD15) or a gene's expression (Myc). The effects of TUG1 and miR-145 on important functional endpoints such as GSC self-renewal, apoptosis, proliferation, or tumorigenesis was not investigated. This should be performed to demonstrate the biological relevance of TUG1 and miR-145 in GSCs.
- 3- What are the relative and contextual contributions of miR-145 and PRC2 to the TUG1 effects? This issue should ideally be ideally experimentally addressed or at least discussed.
- 4- The interactions between TUG1 and mir-145 are presented as if they were an original discovery by the authors. These interactions have been published before (Tan et al., 2015). The authors cite the paper in question but in a completely different context. This issue should be corrected.

Reviewer #3 (Remarks to the Author)

The authors of the manuscript entitled "Targeting the Notch-Regulated Non-Coding RNA, TUG1, as an Effective Treatment in Glioma" take a multifaceted approach to identify lncRNAs regulated in Glioma Stem Cell (GSCs) lines specifically through Notch1 signaling. The authors first characterize the transcriptome in GSCs and then hone in on lncRNAs that have Notch regulatory motifs in their promoter regions and intersect this with Notch1 LOF studies. The intersection leads to a top candidate lncRNA, TUG1. The authors proceed to demonstrate that TUG1 LOF affects GSCs growth (e.g. Loss of SOX2, MYC, Nestin and CD15, apoptosis). They further show that gain of function of a partial TUG1 transcript results in "sponging" of mir-145 and can be rescued by GOF of mir-145. In the nucleus the authors demonstrate that TUG1 localizes to promoters enriched in YY1 sites and are preferentially upregulated upon TUG1 LOF. Moreover, the authors demonstrate similar loss of glioma cancer stem cell repopulation activity in "xenograph" models using oligo mediated TUG1 LOF. Collectively, this study is a tour de force in investigating the role of TUG1 in GSCs. With some important considerations below, this manuscript would be of interest to the general readership of Nature Communications.

1) The authors provide a nice resource of RNA sequencing in GSCs yet identify regulated lncRNAs via microarray analysis. It would improve the resource utility of this manuscript and the impact therein if the authors performed RNA-seq in the Notch1 LOF studies. Thus, the general readership could compare lncRNAs expressed in GSCs with a comprehensive list of those regulated by Notch1, rather than the limited overlap with microarray analyses in LOF studies.

2) Even though the GSC cell lines used have been previously characterized, variations occur with cell culture, especially regarding the self-renewal or repopulating potential of these cells. Since the authors claim their findings support a novel Notch1-lncRNA axis in regulating the self-renewal of glioma cells, they should at least give more details into the tumor initiating potential they were able to obtain with the GSC cell lines and from which the data is derived. The data presented for the limiting dilution assays to assess the glioma stem cell frequencies of the GSC cell lines is insufficient since only the number of cells injected is provided with the mention "validated" (supplementary figure 1C). To appropriately evaluate the assay, the authors should provide the number of mice and the dose for each of them that was injected, the number of mice that developed tumors for each dose and provide the resulting repopulating cell frequency or show the resulting linear regression curve derived from this data.

3) The rescue of Sox2 and Myc expression by injecting TUG1 RNA was done with a truncated Transcript (1-2132bp) (page 5, line 20, figure 2G). Full length TUG1 is a 7.1Kb transcript. How did the authors decide on using this fragment for the rescue experiment? Since the authors already have generated different fragments of TUG1 for other experiments, they should show whether other regions of TUG1 rescue just as well as this 1-2132bp fragment. For example, does the fragment containing only the PRC2 binding region rescue those defects? Or is only the region containing the mir145 site able to rescue? Also, the authors should include a negative control by injecting a random RNA transcript to show that this rescue is specific to the TUG1 fragment expressed and not only due to injection of RNA.

Additionally, the authors should show whether expression of Nestin and CD15 and more importantly if the cell morphological and functional characteristics (just like figure 2C and supplemental figure 3D) and apoptosis is also rescued by injecting TUG1 RNA. Since gamma-secretase inhibitors are not that specific, the authors should also show whether injecting TUG1 RNA can rescue cell growth defects and stem cell marker expression observed when knocking down Notch1 and Jag1 with siRNA.

Additional Considerations:

1) The authors did not provide any indications of the number of replicates performed and the barely enough information about the statistical analyses used for each experiment presented in the manuscript. This is a key piece of information if one wants to assess the quality of the experiments

performed and validity of the results presented.

2) No microarray expression data on the 6 lncRNAs supposedly downregulated after knockdown of Notch1, Jag1 or treatment with DAPT and RO4929097 in the two GSC cell lines is provided (page 4 line 26). Please show the microarray data and the actual downregulation effect for those 6 mentioned lncRNAs. Also, apart from the mention "Data analysis was performed with Gene Spring GX" in the methods section, the authors provide no details about how the data was analyzed (background correction, normalization, statistics, etc.). Finally, although the accession numbers for the microarray data are indicated, providing the microarray results in supplements would also be good practice and make it easier for the readership to access and look at.

3) "These data indicate that Notch signaling predominantly regulates TUG1 expression in glioma, especially in the GSC population" (page 4, last line). For this statement, there is not enough information in the limiting dilution assay to precisely know the frequency of GSC in the cell lines used (a minimum number of 100 cells was injected and gave tumors, so we currently don't know if more than 1% of cells from these cell lines are GSCs capable of generating a tumor). The effect of downregulating the Notch1 pathway on TUG1 is convincing, but most cells may not be a true GSC (until proven otherwise). I would suggest to reformulate the sentence with "These data indicate that Notch signaling predominantly regulates TUG1 expression in glioma cells."

4) Delete the sentence: "These findings may represent an interaction between Notch1 and JAG1, reflective of niche-stem cell communication in clinical GBM tissues (Fig. 6b, c)" (page 5, line 12). Or move to discussion.

Reviewer #1 (Remarks to the Author):

Overall, Reviewer #1 was enthusiastic about the work. We thank the reviewer for acknowledging that “Nevertheless, by using this delivery system, the volume of brain glioma was impressively decreased which makes this paper of interest.” We believe that our findings highlight the potency of *TUG1* targeting as glioma treatment.

Reviewer #2 (Remarks to the Author):

Overall, Reviewer #2 was also enthusiastic about the work. We thank the reviewer for acknowledging that “The therapeutic data are particularly compelling. The work contains complementary mechanistic, expression and experimental therapeutic components and is mostly well performed.” The reviewer had several suggestions. As indicated below, we have taken all the comments and suggestions into account in the revised version of our paper.

1- The data are almost entirely reliant on loss of function (mostly siRNA-based) approaches and should be complemented with gain of function approaches. Specifically, the authors should use Notch gain of function (e.g. NICD overexpression) to show the effects of Notch on TUG1. Also, TUG1 overexpression should be used to assess the effects of TUG1 on molecular and functional endpoints in GSCs.

We agree with this comment. We examined the effects of Notch on *TUG1* expression in two GSCs by NICD overexpressed (Supplementary Fig. 3e-g). We also performed *TUG1* overexpression in two GSCs to assess the effects of *TUG1* on molecular and functional endpoints in GSCs, such as cell growth, apoptosis, and expressions of *SOX2*, *MYC*, *Nestin*, and *CD15* (Fig. 6 and Supplementary Fig. 7). Especially, in order to examine which exons of *TUG1* are important for stemness features, we transfected each exon of *TUG1* separately. This important issue was also raised by Reviewer #3. These data were added in the Figures and Supplementary Figures and discussed in the Discussion section.

2- The functional assays are generally sparse and not fully convincing. With one limited exception (Fig 2B), the readout is a marker (Sox2, nestin, CD15) or a gene's expression (Myc). The effects of TUG1 and miR-145 on important functional endpoints such as GSC self-renewal, apoptosis, proliferation, or tumorigenesis was not investigated. This should be performed to demonstrate the biological relevance of TUG1 and miR-145 in GSCs.

In addition to *TUG1* loss of function approaches (Fig. 2, Fig. 7 and Fig. 8), we performed *TUG1* overexpression in GSCs and showed the important functional endpoints (e.g. stemness markers,

apoptosis, proliferation) of GSCs (Fig. 6 and Supplementary Fig. 7). Regarding miR-145, we also examined the biological relevance of miR-145 on important functional endpoints by induction of a precursor molecule mimicking miR-145 in GSCs (Supplementary Fig. 5). These data were added in the Figures and Supplementary Figures and discussed in the Discussion section.

3- What are the relative and contextual contributions of miR-145 and PRC2 to the TUG1 effects? This issue should ideally be ideally experimentally addressed or at least discussed.

This is an important point. In order to examine which regions of *TUG1* are important for stemness features, we transfected each exon of *TUG1*. *TUG1* exon 1 and exon 2 function as a miR-145 sponge and PRC2 recruitment to a certain loci, respectively. We found that *TUG1* exon 1 is an essential region to maintain stemness (Fig. 6 and Supplementary Fig. 7). These data were added in the Figures and Supplementary Figures and discussed in the Discussion section.

4- The interactions between TUG1 and mir-145 are presented as if they were an original discovery by the authors. These interactions have been published before (Tan et al., 2015). The authors cite the paper in question but in a completely different context. This issue should be corrected.

We sincerely apologize for not providing appropriate information regarding the previous study. We corrected the text.

Reviewer #3 (Remarks to the Author):

We are grateful to the reviewer for acknowledging that “this study is a tour de force in investigating the role of *TUG1* in GSCs” and for his/her thoughtful critique. The reviewer raises several important points for further consideration.

1) The authors provide a nice resource of RNA sequencing in GSCs yet identify regulated lncRNAs via microarray analysis. It would improve the resource utility of this manuscript and the impact therein if the authors performed RNA-seq in the Notch1 LOF studies. Thus, the general readership could compare lncRNAs expressed in GSCs with a comprehensive list of those regulated by Notch1, rather than the limited overlap with microarray analyses in LOF studies.

We agree with this comment. We performed RNA-seq analysis in two GSCs treated with either

Notch1-siRNA or *JAG1*-siRNA (Supplementary Fig. 2 and Supplementary Table 2). Data were quite consistent between lncRNA microarray and RNA-seq analyses. These data, including a list of lncRNAs regulated by Notch1, were added to the Supplementary Figures and Supplementary Tables.

2) *Even though the GSC cell lines used have been previously characterized, variations occur with cell culture, especially regarding the self-renewal or repopulating potential of these cells. Since the authors claim their findings support a novel Notch1-lncRNA axis in regulating the self-renewal of glioma cells, they should at least give more details into the tumor initiating potential they were able to obtain with the GSC cell lines and from which the data is derived. The data presented for the limiting dilution assays to assess the glioma stem cell frequencies of the GSC cell lines is insufficient since only the number of cells injected is provided with the mention "validated" (supplementary figure 1C). To appropriately evaluate the assay, the authors should provide the number of mice and the dose for each of them that was injected, the number of mice that developed tumors for each dose and provide the resulting repopulating cell frequency or show the resulting linear regression curve derived from this data.*

This is an important point. Although we characterized our GSCs and showed data from our previous study, we further validated our GSCs to show more details about the self-renewal and tumor initiating potential. We performed limiting dilution assays and found that even 100 cells of each GSC line, which we used in the current study, were sufficient to establish tumors with 100% (Supplementary Fig. 1c). These data are consistent with previous work (Singh, S.K., *et al.* Identification of human brain tumour initiating cells. *Nature* **432**, 396-401 (2004).). We also added this reference in the text.

3) *The rescue of Sox2 and Myc expression by injecting TUG1 RNA was done with a truncated Transcript (1-2132bp) (page 5, line 20, figure 2G). Full length TUG1 is a 7.1Kb transcript. How did the authors decide on using this fragment for the rescue experiment? Since the authors already have generated different fragments of TUG1 for other experiments, they should show whether other regions of TUG1 rescue just as well as this 1-2132bp fragment. For example, does the fragment containing only the PRC2 binding region rescue those defects? Or is only the region containing the mir145 site able to rescue? Also, the authors should include a negative control by injecting a random RNA transcript to show that this rescue is specific to the TUG1 fragment expressed and not only due to injection of RNA.*

Additionally, the authors should show whether expression of Nestin and CD15 and more importantly if the cell morphological and functional characteristics (just like figure 2C and

supplemental figure 3D) and apoptosis is also rescued by injecting TUG1 RNA. Since gamma-secretase inhibitors are not that specific, the authors should also show whether injecting TUG1 RNA can rescue cell growth defects and stem cell marker expression observed when knocking down Notch1 and Jag1 with siRNA.

Thank you for this comment. The transcript (1-2132bp corresponding to *TUG1* exon 1) contains miR-145-binding sites and acts as a miRNA sponge. The *TUG1* Exon 2 (2133-2910bp) binds to PRC2 and YY1. Function of *TUG1* exon 3 is still unknown. In the revised version, we transfected each exon of *TUG1* to examine which of the *TUG1* exons is essential for the stemness features (Fig. 6 and Supplementary Fig. 7). We found that injection of *TUG1* Exon 1 rescued the expression of *Sox2*, *Myc*, *Nestin*, and *CD15*, cell growth defects, and apoptosis in Notch-inhibited GSCs (i.e. knocking down *Notch1* and *Jag1* with siRNA or gamma-secretase inhibitors). These data were added in the Figures and Supplementary Figures and discussed in the Discussion section. Regarding the negative control by injecting a random RNA transcript, we added these control experiments in the revised version.

Additional Considerations:

1) The authors did not provide any indications of the number of replicates performed and the barely enough information about the statistical analyses used for each experiment presented in the manuscript. This is a key piece of information if one wants to assess the quality of the experiments performed and validity of the results presented.

We sincerely apologize for not providing greater details regarding the experimental replicates and statistical analyses. We added such information to the Materials and Methods.

2) No microarray expression data on the 6 lncRNAs supposedly downregulated after knockdown of Nothc1, Jag1 or treatment with DAPT and RO4929097 in the two GSC cell lines is provided (page 4 line 26). Please show the microarray data and the actual downregulation effect for those 6 mentioned lncRNAs. Also, apart from the mention "Data analysis was performed with Gene Spring GX" in the methods section, the authors provide no details about how the data was analyzed (background correction, normalization, statistics, etc..). Finally, although the accession numbers for the microarray data are indicated, providing the microarray results in supplements would also be good practice and make it easier for the readership to access and look at.

We agree with this comment. We showed the microarray data and the actual downregulation

effect for those 6 mentioned lncRNAs (Supplementary Table 3). We apologize for not providing greater details regarding the analysis of microarray data. We added this information in Materials and Methods. We also added the microarray results to the Supplemental Tables.

3) *"These data indicate that Notch signaling predominantly regulates TUG1 expression in glioma, especially in the GSC population" (page 4, last line). For this statement, there is not enough information in the limiting dilution assay to precisely know the frequency of GSC in the cell lines used (a minimum number of 100 cells was injected and gave tumors, so we currently don't know if more than 1% of cells from these cell lines are GSCs capable of generating a tumor). The effect of downregulating the Notch1 pathway on TUG1 is convincing, but most cells may not be a true GSC (until proven otherwise). I would suggest to reformulate the sentence with "These data indicate that Notch signaling predominantly regulates TUG1 expression in glioma cells."*

Thank you for this comment. We corrected the sentence as "These data indicate that Notch signaling predominantly regulates *TUG1* expression in glioma cells."

4) *Delete the sentence: "These findings may represent an interaction between Notch1 and JAG1, reflective of niche-stem cell communication in clinical GBM tissues (Fig. 6b, c)" (page 5, line 12). Or move to discussion.*

Thank you for this comment. We deleted the sentence.